

**HESS Opinions: Chemical transport modeling in subsurface hydrological**
**systems – Space, time, and the holy grail of "upscaling"**

Brian Berkowitz[1]
[1]Department of Earth and Planetary Sciences, Weizmann Institute of Science, Rehovot 7610001,
Israel
*Correspondence to:* Brian Berkowitz (brian.berkowitz@weizmann.ac.il)
**Abstract.**
Extensive efforts over decades have focused on quantifying chemical transport in subsurface
geological formations, from microfluidic laboratory cells to aquifer field scales. Outcomes of these
efforts have remained largely unsatisfactory, however, largely because domain heterogeneity (in
terms of, e.g., porosity, hydraulic conductivity, geochemical properties) is present over multiple
length scales, and "unresolved", practically unmeasurable heterogeneities and preferential
pathways arise at virtually every scale. While *spatial* averaging approaches are effective when
considering overall fluid flow – wherein pressure propagation is essentially instantaneous and the
system is "well mixed" – purely *spatial* averaging approaches are far less effective for chemical
transport, essentially because well-mixed conditions do not prevail. We assert here that an explicit
accounting of *temporal* information, under uncertainty, is an additional, but *fundamental*,
component in an effective modeling formulation. As an outcome, we further assert that "upscaling"
of chemical transport equations – in the sense of attempting to develop and apply chemical
transport equations at large (length) scales, based on measurements and model parameter values
obtained at significantly smaller length scales – is very much a holy grail. Rather, we maintain that
it is necessary to formulate, calibrate and apply models using measurements at similar scales of
interest, in both space and time.
**Keywords:** Preferential flow, anomalous transport, numerical modeling, measurements





## 1 Introduction

### 1.1 Background

There have been extensive efforts over the last ~60 years to model and otherwise quantify fluid flow and chemical (contaminant) transport in soil layers and subsurface geological formations, from millimeter-size, laboratory microfluidics cells to aquifer field scales extending to hundreds of meters and even tens of kilometers.

Soil layers and subsurface formations can exhibit significant heterogeneity, in terms of domain characteristics such as porosity, hydraulic conductivity, structure, and biogeochemical properties (mineral and organic matter content). However, recognition that effects of heterogeneity over multiple length scales, with "unresolved", practically unmeasurable heterogeneities arising at every length scale (from pore to field) cannot be simply "averaged out", has become broadly accepted only more recently. Indeed, much research on flow and transport in porous media, dating particularly from ~1950-1990, but also essentially to date, has been based on the search for length scales at which one can define a "representative elementary volume", or otherwise-named "averaging volume", above which variability in fluid and chemical properties become constant. In this context, too, many varieties of homogenization, volume averaging, effective medium, and stochastic continuum theories have been developed in an extensive literature. These methods allowed formulation of continuum-scale, generally Eulerian, partial differential equations to quantify ("model") fluid flow and chemical transport, which were then applied in the soil and groundwater literature at length scales ranging from millimeters to full aquifers. While originally deterministic in character, a variety of stochastic formulations, and use of Monte Carlo numerical simulation techniques, introduced from the 1980s, enabled analysis of uncertainties in input parameters such as hydraulic conductivity.

However, while analysis of fluid flow using these methods has proven relatively effective, quantification of chemical transport, and an accounting of associated (biogeo)chemical reactions in cases of reactive chemical species and/or host porous media, has remained largely unsatisfactory. We discuss the reasons for this, in detail, in the sections below. Briefly, the overarching reason for these successes and failures is that *spatial* averaging approaches are effective when considering overall fluid flow rates and quantities, wherein pressure propagation is essentially instantaneous and the system is "well mixed" (because mixing of water "parcels" is functionally irrelevant). However, purely *spatial* averaging approaches are far less effective for chemical transport, essentially because well-mixed conditions do not prevail, and spatial averaging is inadequate; here, an explicit, additional accounting of *temporal* effects is required.

The focus of the current contribution is on modeling conservative chemical transport in geological media. In terms of modeling, one can delineate two main types of scenarios: (i) *pore-scale modeling* in relatively small domains, with a detailed (and specified) pore structure, and (ii) *continuum-scale modeling* in porous media domains, that average pore space and solid phases at scales from laboratory flow cells to field-scale plots and aquifers. Case (i) requires, e.g., Navier-





Stokes or Stokes equations solutions for the underlying flow field, coupled with solution of a local
(e.g., advection-*diffusion*) equation for transport, while Case (ii) requires Darcy (or related)
equation solutions for the underlying flow field, coupled with solution of a governing transport
equation for chemical transport. *Note:* here and throughout, we shall use the terms "*continuum*
*level*" and "*continuum scale*" in reference to case (ii) scenarios, and "*pore-scale*" to refer to Case
(i) scenarios, although we recognize, too, that pore-scale Navier-Stokes and advection-diffusion
equations are continuum partial differential equations.

***Disclaimer:*** Here and throughout this contribution, the overview comments and references to
existing philosophies, methodologies and interpretations are written, largely, in broad terms,
without (necessarily limited numbers of) citations selected from the vast literature. This approach
is taken with a clear recognition and respect for the body of literature that has driven our field
forward over the last decades, but with the express desire to avoid any risk of unintentionally
alienating colleagues and/or misrepresenting aspects of relevant studies. *As an Opinion*
*contribution, and with length considerations in mind, there is no attempt to provide an exhaustive*
*listing and description of relevant literature.*

**1.2 Assertions**

The pioneering paper of Gelhar and Axness (1983) focused on quantifying conservative chemical
transport at the continuum level. They expressed heterogeneity-induced chemical spreading in
terms of the (longitudinal) macrodispersion coefficient – as it appears in the classical
(macroscopically 1d) advection-dispersion equation – with knowledge of the variance and
correlation length of the log-hydraulic conductivity field and the mean, ensemble-averaged fluid
velocity. The conceptual approach embodied in Gelhar and Axness (1983) – and by many
researchers since then (as well as previously) – was founded on delineation of the *spatial*
*distribution* of the hydraulic conductivity and application of an averaging method to determine the
governing transport equation with "effective parameters" to describe chemical transport at a given
length scale.
In contrast, we assert here that *spatial* information, *alone*, is generally *insufficient* for
quantification of chemical transport phenomena. Rather, *temporal* information is an additional,
but *fundamental*, component in an effective modeling formulation. In the discussion below, we
shall justify this argument by a series of examples. We examine (i) *spatial* information on, e.g.,
the hydraulic conductivity distribution at the continuum level, or distribution of the solid phase at
the pore-scale level; and (ii) *temporal* information on, e.g., contaminant (tracer, "particle")
transport mobility and retention in different regions of a domain. We thus define a type of
"information hierarchy", with different types of information required for different flow and
chemical transport problems of interest.
As an outcome of the above assertion and the discussion below, we further assert that
"upscaling" of chemical transport equations – development and application of chemical transport


equations at large (length) scales, with corresponding parameter values, based on measurements
and model parameter values obtained at significantly smaller length scales – is very much a holy
grail. Rather, we maintain that it is necessary to formulate, calibrate and apply models using
measurements at similar scales of interest, in both space and time. This does not exclude use of
similar equation *formulations* at different spatial scales, but it does entail use of different parameter
values, at the relevant scale of interest, that cannot be determined a priori or from purely spatial or
flow-only measurements.

**1.3 Approach – Outline**

While our focus is on chemical transport, knowledge of fluid flow and delineation of the velocity
field throughout the domain is a prerequisite. We therefore first discuss fluid flow as an intrinsic
element aspect of the "information hierarchy". Specifically, we address how:

(1) Basic structural information on "conducting elements" in a system representing a geological domain (porous and/or fractured) can provide basic insight regarding overall fluid conduction in the domain, as a function of "conducting element" density. We emphasize that without direct simulation of fluid flow (as well as chemical transport) in such a system, this type of analysis is insufficient in terms of defining the actual flow field and velocity distributions throughout the domain.

(2) Spatial information on, in particular, the hydraulic conductivity distribution at a continuum scale, or solid phase distribution at the pore scale, throughout the domain, can be used to *determine the flow field*. We then show that this is insufficient in terms of defining chemical transport.

(3) Temporal information on chemical species migration, which quantifies distributions of retention and release times (or rates) of chemicals by advective-dispersive-diffusive and/or chemical mechanisms, can be used to *determine the full spatial and temporal evolution of a migrating chemical plume*, either by solution of a transport equation or use of particle tracking on the velocity field.

We comment, parenthetically, that in conceptual-philosophical terms, this hierarchy and the "limitations" of each level are in a sense analogous to representation of geometrical constructs in multiple dimensions: in principle, one can represent, as a *projection*, a d-dimensional object in d–1 dimensions. But of course, by its very nature, a *projection* does not capture all features of the construct in its "full" dimension. To illustrate, an (imaginary) 1d curve can represent a 2d Möbius strip, a 2d perspective drawing can represent a 3d cube, and a 3d construct can represent a 4d object (where the 4[th] dimension might be considered time) — and yet, none of these d–1 dimensional representations contains all features of the actual d-dimensional objects. Similarly, despite our frequent attempts to the contrary, one cannot properly describe (2) only from (1), or (3) only from (2).



## 2 Fluid flow

Prior to actually solving for fluid flow, to determine the underlying velocity field, efforts are sometimes invested in considering geometrical (structural) information, for example, when examining fracture networks in essentially impermeable host rock.

In this context, percolation theory (Stauffer and Aharony, 1994) is particularly useful in determining, statistically, whether or not a domain with $N$ "conducting elements" (e.g., fractures) is includes sufficient element density to form a connected pathway enabling fluid flow across the domain. One can estimate, in this context, the critical value, $N_c$, for which the domain is "just" connected, as a function of fracture length distribution, or the critical average fracture length as a function of $N$ needed to reach domain connectivity (Berkowitz, 1995). Similarly, percolation theory shows how the overall hydraulic conductivity of the domain scales as the number of conducting elements, $N$, relative to the $N_c$ critical number of conducting elements required for the system to begin to conduct fluid. Percolation theory also addresses diffusivity scaling behavior of chemical species. But, fundamentally, percolation is a statistical framework suitable for large ("infinite") domains, and provides universal scaling behaviors with no coefficient of equality.

Other approaches have been advanced to analyze domain connectivity, e.g., using graph theory and concepts of identification of paths of least resistance in porous medium domains (e.g., Rizzo and de Barros, 2017). Like percolation theory, such approaches provide useful information and "estimates" on the hydraulic connectivity and flow field, and even on first arrival times of chemical species, without solving equations for fluid flow and chemical transport. However, these methods to not provide full delineation of the flow field and velocity distribution throughout a domain.

It is thus clear that, in general, there are dynamic aspects of fluid flow, over and above pure structure: knowledge of pure geometry is not sufficient, and *we must actually solve for the flow field*, at either the pore-scale or a continuum scale, to determine the velocity field and actual flow paths throughout the domain. Delineation of a flow field and velocity distribution by solution of the Navier-Stokes equations (or Stokes equation for small Reynolds numbers), or by solution of the Darcy equation, may be considered "rigorous", correct and effective. But in the process of solving for the flow field, two *key* features arise, one more relevant to pore-scale analyses, and the other more relevant to continuum-scale analysis, as detailed in Sect. 2.1 and Sect. 2.2, respectively.

### 2.1 Pore-scale flow field analysis

Why is knowledge only of the geometrical "static" structure (spatial distribution of solid phase) insufficient to know the flow dynamics in a pore-scale domain? Consider the 2d domain shown in Figure 1, containing sparsely and randomly distributed obstacles (porosity of 0.9). Figure 1 shows solutions of the Navier-Stokes equations for two Reynolds number (Re) values. [Recall: $\mathrm{Re} \equiv \rho v L / \mu$, where $\rho$ and $\mu$ are density and dynamic viscosity of the fluid, respectively, $v$ is fluid velocity, and $L$ is a characteristic linear dimension.]. Andrade et al. (1999) showed clearly that





well-defined preferential flow channels at lower Re, while at higher Re, channeling is less intense
and the streamline distribution is more spatially homogeneous in the direction orthogonal to the
main flow.
Figure 1 demonstrates that the streamlines in individual pores change because of the interplay
between inertial and viscous forces, given by Re. In other words, with a change in overall fluid
velocity (or hydraulic gradient) across the domain, the actual flow paths can be altered, together
with a change in overall and (spatially) local residence times of fluid molecules (and chemical
species, as addressed below). Of course, the significantly lower porosities and more tortuous pore
space configuration in natural, heterogeneous geological porous media may affect the impact of
inertial effects, but the principle remains relevant. [We note, too, parenthetically, that the behavior
shown in Fig. 1 is relevant also to fluid flow within fracture planes, wherein the obstacles represent
contact areas and regions of variable aperture.]
Clearly, then, except in highly idealized and simplified geometries, use of a purely analytical
solution to identify the full velocity field and streamline patterns is not feasible. Moreover, the
extent and changes in streamlines are not intuitively obvious without full numerical solution of the
governing flow equations, for any specific set of porous medium structures and boundary
conditions.

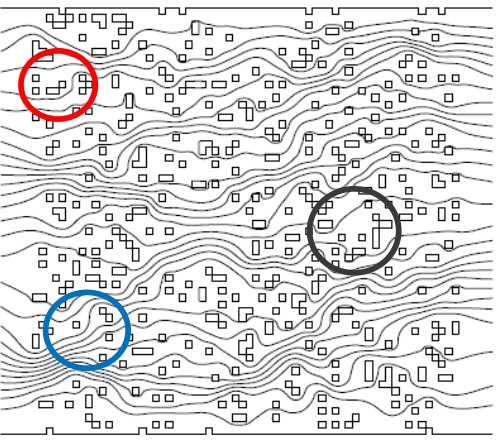 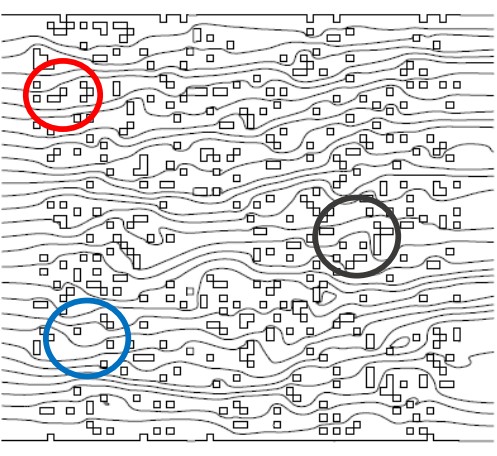

(a) Re = 0.0156                           (b) Re = 15.6

**Figure 1.** 2d domain containing randomly distributed obstacles (squares and rectangles). Stream functions for (a) Re = 0.0156 and (b) Re = 15.6 are shown with constant increments between consecutive streamlines (modified from Andrade et al., 1999, https://doi.org/10.1103/PhysRevLett.82.5249; Copyright, American Physical Society). The different patterns of preferential pathways are clear and distinct. The three pairs of circles (red, blue, black) highlight three (of many) specific locations where the streamlines are seen to change as a function of Re.





## 2.2 Continuum-scale flow field analysis

Considering now continuum-scale domains, but in analogy to the example shown in Sect. 2.1, we illustrate why knowledge only of the geometrical "static" structure (spatial distribution of the hydraulic conductivity) is insufficient to know the flow dynamics, without solution of the Darcy equation.

Figure 2 represents a realization of a numerically-generated (statistically homogeneous, isotropic, Gaussian) hydraulic conductivity ($K$) 2d domain. The Darcy equation solution for this domain yields values of hydraulic head throughout the domain; these are converted to local velocities, to enable delineation of the streamlines and preferential flow paths. The latter are highlighted by actually solving for chemical transport, by following the migration of "particles" representative of masses of dissolved chemical species injected along the inlet boundary of the flow domain; see Edery et al. (2014) for details. Of particular significance is that 99.9% of the injected particles travel in preferential pathways through a limited number of domain cells. We return to Figure 2 in Sect. 3.3.2, where we discuss a framework that effectively characterizes and quantifies chemical transport.

Unlike the pore-scale case shown in Sect. 2.1, at the Darcy/continuum scale, streamlines are not altered with changes in the overall hydraulic gradient, as long as laminar flow conditions are maintained. And yet, preferential flow paths are (surprisingly) sparse and ramified, sampling only limited regions of a given heterogeneous domain, with the vast fraction of a migrating chemical species that interrogates the domain being even more limited. Significantly, except in highly idealized and simplified geometries (e.g., homogeneous media), delineation of these pathways is not intuitively obvious (e.g., by simple inspection of the hydraulic conductivity map in Fig. 2a) or definable from a priori analysis or tractable analytical solution. Rather, numerical solution of the governing flow equations is required, for any particular/specific set of porous medium structures and boundary conditions. [Note, too, that critical path analysis from percolation theory (discussed in Sect. 2) – again from purely "static" information without solution of the flow field – yields an incorrect interpretation, as shown in detail by Edery et al. (2014).]

We emphasize that the delineation of "preferential flow paths" is usually relevant only for study of chemical transport; if water *quantity*, alone, is the focus, then specific "flow paths" travelled by water molecules – and their advective and diffusive migration along and between streamlines, and into/out of less mobile regions – are of little practical interest. The movement of chemical species, on the other hand, which experience similar advective and diffusive, and thus "dispersive", transfers, must be monitored closely to be able to quantify overall migration through a domain. We return to consider patterns of chemical migration in Sect. 3. But this argument, too, reinforces the assertion that delineation of actual chemical transport cannot be deduced purely from spatial information and solution for fluid flow, but must be treated by solution of a transport equation.

(a)

(c)

(d)

**Figure 2.** Maps of (a) hydraulic conductivity, $K$, distribution in a domain with $300 \times 120$ cells, (b) preferential pathways for fluid flow (and chemical transport), and (c) preferential pathways through cells that each contain a visitation of at least 0.1% of the total number of chemical species particles injected into the domain (flux-weighted, along the entire inlet boundary). Flow is from left to right. Note that the color bars are in $\ln(K)$ scale for Figure 2a, and $\log_{10}$ number of particles for Figures 2b,c (modified from Edery et al., 2014; © with permission from the American Geophysical Union 2014). (d) Laboratory flow cell, 2.13 m length, with an exponentially correlated K structure, showing preferential pathways for blue dye injected near the inlet (flow is left to right); dark, medium, and light colored sands represent high, medium and low conductivity, respectively (modified from Levy and Berkowitz, 2003; © with permission from Elsevier 2003). The circles shown in (c) and (d) highlight two (of many) regions in which the pathways are seen to contain lower $K$ "bottlenecks".

It is significant, too, that fluid flow (and chemical transport) occurs in preferential pathways that contain low conductivity sections (indicated by circles in Figs. 2c,d). How do we explain passage through "bottlenecks" (low hydraulic conductivity patches) within the preferential





pathways, and that fluid (and chemicals) do not migrate "only" through the highest conductivity
patches?
To address this question, we begin by considering what happens in a 1d path. Consider two
paths, each containing a series of five porous medium elements (or blocks), with distinct hydraulic
conductivity ($K_i$) values. Consider Path 1, with a series hydraulic conductivity values of 3, 3, 3, 3,
3, and Path 2, with values 6, 6, 1, 6, 6 (specific length/time units are irrelevant here). The value of
$K = 1$ represents a clear "bottleneck" in an otherwise higher $K$ path than that of Path 1. In a 1d
series, however, the overall hydraulic conductivity ($K_{overall}$) of the path is given by the harmonic
mean of the conductivities of the elements comprising the path: $K_{overall} = 5 / (\Sigma_{i=1,5} 1/K_i)$;
significantly, in the two cases here, both paths have $K_{overall} = 3$! So a "bottleneck" ($K=1$) can be
"overcome" and does not cause necessarily cause a potential pathway to be less "desirable" than a
pathway without such "bottlenecks". Of course, in 2d and 3d systems, patterns of heterogeneity
and pathway "selection" by water/chemicals are significantly more "complicated", but the
principle discussed here for 1d systems still holds, in the sense that lower hydraulic conductivity
("bottleneck") elements can (and do) exist in the preferential pathways (Margolin et al., 1998;
Bianchi et al., 2011).
**3 Chemical transport**
We now consider the next level of the "information hierarchy" outlined in Sect. 1.3. To quantify
the evolution of a migrating chemical plume, knowledge of the flow field is not generally
sufficient, and additional means to characterize and quantify the behavior are needed. Dynamic
aspects of chemical transport require us to think (also) in terms of *time*, not just *space* and physical
structure. Moreover, it is generally insufficient to determine the transport of the chemical plume
center of mass. Rather, in terms of water resource contamination and remediation, for example, it
is critical to characterize, respectively, the early and late arrival times at compliance (monitoring)
regions downstream of the region (point, areal, volumetric) in which the chemical species entered
the system.
As we show below, it becomes clear that, in general, there are dynamic aspects of chemical
transport, on over and above the role of the flow field, and *we must actually solve for chemical*
*transport*, at either the pore-scale or a continuum scale, to determine the spatiotemporal (spatial
plume and/or temporal breakthrough curve) evolution of the migrating chemical plume. In *both*
pore-scale and continuum-scale domains, the *critical* control that arises is that of time (in addition
to space). This is in sharp contrast to fluid flow at pore and continuum scales, as shown in Sect.
2.1 and Sect. 2.2: pore-scale fluid flow displays changing streamlines with changes in hydraulic
gradient, while continuum-scale fluid flow follows distinct but difficult to identify preferential
flow paths essentially independent of the hydraulic gradient.
We point out, too, that for both pore-scale and continuum-level scenarios, one can solve,
explicitly, a governing equation for transport. Alternatively, though, one can obtain an
"equivalent" solution by solving for (Lagrangian framework) "particle tracking" of transport along





the calculated streamlines. In other words, particle tracking methods essentially represent an
alternative means to solve an ((integro-)partial differential) equation for chemical transport; such
methods can be applied, too, when the precise partial differential equation is unknown or the
subject of debate. We also note that solution of the relevant equations for fluid flow and chemical
transport is sometimes achieved by (semi-)analytical methods, if the flow/transport system can be
treated sufficiently simply (e.g., macroscopically, section-averaged 1d flow and transport in a
rectangular domain).
We first discuss principal features of pore-scale (Sect. 3.1) and continuum-scale (Sect. 3.2)
chemical transport, and in Sect. 3.3, we focus on effective model formulations.

**3.1 Pore-scale chemical transport analysis**

To illustrate why knowledge only of only the flow field is insufficient for full quantification of
chemical transport, consider the three porous medium domains shown in Fig. 3. Each domains is
comprised of pore-scale images of a natural rock, modified by enlarging the solid phase grains, to
yield three different configurations: a statistically homogeneous system domain, a weakly
correlated system, and a structured, strongly correlated system (see Nissan and Berkowitz (2019)
for details). Fluid flow was determined by solution of the Navier-Stokes equations (Fig. 1a).
Transport of a conservative chemical species was then simulated via a (Lagrangian) streamline
particle tracking method, for an ensemble of particles that advance according to a Langevin
equation. Transport behavior was determined for two values of macroscopic (domain average)
Péclet number (Pe). [Recall; Pe $\equiv vL/D$, where $v$ is fluid velocity, $L$ is a characteristic linear
dimension, and $D$ is the coefficient of molecular diffusion.] Here, the macroscopic Pe is based on
the mean particle velocity and mean particle displacement distance per transition ("step").
Figure 3 shows that regardless of possible (pore-scale) streamline changes as a function of
hydraulic gradient (recall Sect. 2.1, considering different values of Re), the choice of *macroscopic*
Péclet number in a given domain plays a significant role in the evolution of the migrating chemical
plume. In particular, the relative effects of advection and diffusion, which vary locally in space,
are critical, as is the overall residence time in the domain. We stress here (and return to this key
point in discussion below) that the spatially (and in some case temporally) *local* changes in relative
effects of advection and diffusion – characterized by the *local* Pe – dominates determination of the
plume evolution. This can be understood from study of Fig. 3, in each of the three heterogeneity
configurations, for two choices of macroscopic Pe values; the different patterns of longitudinal
and transverse spreading are observed clearly.
The behavior show in Fig. 3 is essentially well-known from extensive simulations and
experiments appearing in the literature. This behavior is described here to stress the importance of
*temporal* effects, and to point out that information only of the advective velocity field – as
discussed in Sect. 2.1 and Sect. 2.2 – is not sufficient to "predict" chemical transport.



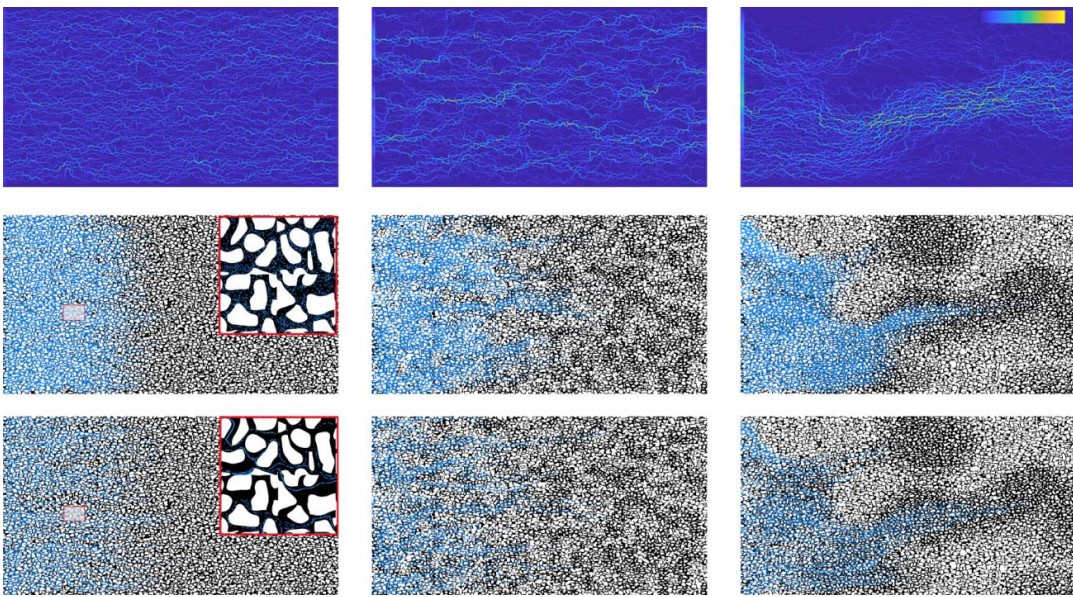

**Figure 3.** Fluid velocities and chemical migration in three porous media configurations (from left to right): homogeneous system, randomly heterogeneous system, and structured heterogeneous system. The upper row shows the (normalized) velocity field for the three configurations; the color bar represents relative velocity, with dark blue being lowest. The middle and lower rows show, respectively, numerically-simulated particle tracking patterns of an inert chemical species (blue dots) at $Pe = 1$ (middle row) and $Pe = 100$ (lower row) for the three configurations (white color indicates solid phase; black color indicates liquid phase). Note: The particles plumes are shown at 10% of the final time of each simulation; absolute travel times differ among the plots. The insets in the left side plots of the middle and lower rows show the pore-scale chemical species distributions; note the more diffuse pattern for $Pe = 1$ (from Nissan and Berkowitz, 2019, https://doi.org/10.1103/PhysRevE.99.033108; © with permission from American Physical Society 2019).

## 3.2 Continuum-scale chemical transport analysis

The aspects discussed in Sect. 3.1 are relevant, analogous and essentially applicable also to chemical transport at the continuum scale. Consider the two laboratory experiments shown in Fig. 4 and Fig. 5. Each flow cell was filled with a different clean, sieved sand configuration; see Levy and Berkowitz (2003) for details. Figure 4 shows a uniform ("homogeneous") packing of clean sand, while Fig. 5 shows a "coarse" sand containing a randomly heterogeneous arrangement of rectangular inclusions consisting of a "fine" sand. The flow cells, fully saturated with water, enabled macroscopically (section-averaged) 1d, steady-state flow, with a mean gradient parallel to the horizontal axis of the cell. As seen in the two figures, neutrally-buoyant, inert red dye was injected at seven (Fig. 4) and five (Fig. 5) points near the inlet side, to illustrate the spatiotemporal evolution of the chemical plumes.

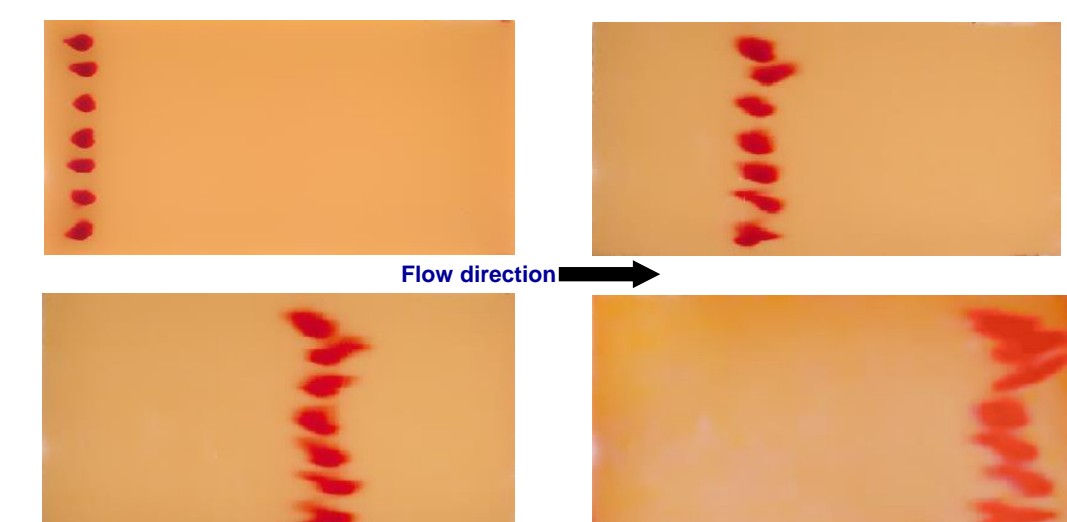

**Figure 4.** Photographs of dye transport in a flow cell (internal dimensions $0.86 \times 0.45 \times 0.10$ m) containing a uniform packing of quartz sand (average grain diameter 0.532 mm), under a constant flow rate, at four times (modified from Levy and Berkowitz, 2003; © with permission from Elsevier 2003).

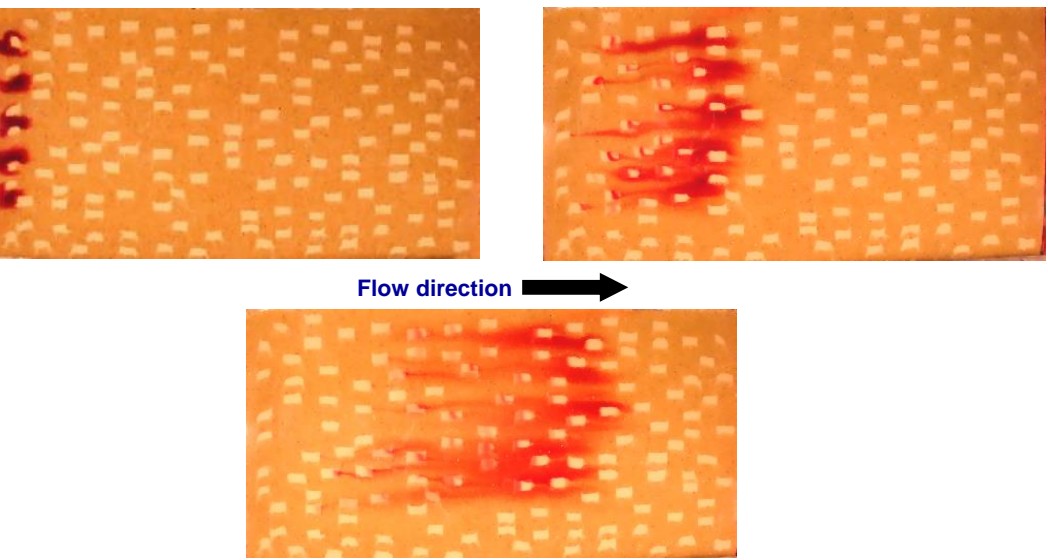

**Figure 5.** Photographs of dye transport in a flow cell (internal dimensions $0.86 \times 0.45 \times 0.10$ m) containing a randomly heterogeneous packing of quartz sand, under a constant flow rate, at three times. The rectangular inclusions comprise sand with an average grain diameter ~0.5× smaller, and hydraulic conductivity ~3× lower, than the surrounding sand matrix (modified from Levy and Berkowitz, 2003; © with permission from Elsevier 2003).



Most notably, in both Fig. 4 and Fig. 5: (i) each of the plumes has a different, unique pattern,
which continues over the duration of the plume migration; and (ii) none of the plumes is
"elliptical", as expected in classical Fickian transport theory and embodied in solutions of the
classical advection-dispersion equation (ADE). Indeed, vertical averaging of each plume shown in
Fig. 4 and Fig. 5, at each time, does not yield Gaussian (normally distributed) concentration
profiles, but rather asymmetrical, "heavy-tailed" profiles.
At this juncture, we note that here and below that we use the terms "non-Fickian", or
"anomalous" – others sometimes use the terms "pre-asymptotic" or "pre-ergodic" – to denote any
chemical transport behavior that differs from that described by the classical ADE or similar type
of continuum-scale formulation. Typically, though, non-Fickian transport is characterized by early
and or late arrival times of migrating chemical species to some control or measurement plane/point,
relative to those resulting from solution of the ADE. The ADE applies to so-called Fickian
behavior, in the sense that it accounts for mechanical dispersion as a macroscopic form of Fick's
law; mechanical dispersion arises as an "effective" (or "average") quantity that describe local
fluctuations around the average (advective) fluid velocity. Thus, in this formulation, a pulse of
chemical introduced into a macroscopically 1d, uniform velocity, for example, leads to temporal
and spatial concentration distributions that are equivalent to a normal (Gaussian) distribution.
It is in this context that the term "homogeneous" packing used above is placed in quotation
marks, to indicate that in natural geological media, "homogeneity" does not really exist. Any
natural geological sample of porous medium contains multiple scales of heterogeneity; and at each
particular scale of measurement, "unresolved" heterogeneities that are essentially unmeasurable
are present (even scanning electron microscopy and atomic force microscopy have limits of spatial
resolution, for example!). And thus, as seen in Fig. 4 for example, the overall transport pattern
even in an "homogeneous" system can be non-Fickian (anomalous). *We therefore emphasize that*
*because natural heterogeneity in geological formations occurs over a broad range of scales,*
*"normal" (Fickian) transport tends to be the "anomaly", whereas "anomalous" (non-Fickian)*
*transport is ubiquitous, and should be considered "normal".*
Moreover, as noted in Sect. 2.2, streamlines are not altered with changes in the overall
hydraulic gradient, at the continuum (Darcy) scale, as long as laminar flow conditions are
maintained, because increasing the hydraulic gradient increases the fluid velocity along the
existing ("predefined") streamlines by the same factor. However, the character of chemical
transport can be altered, as the change in residence time in the domain affects the relative effects
of advection and diffusion space. And in domains with heterogeneous distributions of hydraulic
conductivity, the local Pe (Sect. 3.1) can vary more strongly, too.
Thus, we argue that patterns of chemical transport cannot be fully determined from information
only on the velocity field; solution of an appropriate continuum-scale transport equation cannot be
avoided. In conclusion, then, and with particular reference to the (conceptually and theoretically
beautiful) classical ADE and "conventional" conceptual understanding and quantitative
description of chemical transport, we suggest that one must separate mathematical convenience
and wishful thinking from the reality of experiments: there is a definitive need for more powerful





formulations of transport equations. In this context, one is reminded of the quotation by the
biologist Thomas Henry Huxley: "The great tragedy of science—the slaying of a beautiful theory
by an ugly fact." (*President's Address to the British Association for the Advancement of Science,*
*Liverpool Meeting, 14 Sep 1870*).
**3.3 Modeling chemical transport, and the myth that "fewer parameters is always better"**
So how *do* we effectively model chemical transport?
As noted at the outset of Sect. 2, solution of the Navier-Stokes or Darcy equations to determine
the full *flow* field and velocity distribution in a given porous medium domain has been proven
correct and effective in most applications, and is well-accepted in the literature. However,
modeling of chemical transport is more contentious, the reasons for which we expand upon below.
We argue here that modeling of chemical species transport requires us to think in terms of *time*,
not just *space*. To assist the reader to enter this frame of thinking, and to sharpen our
conceptualization, we provide two examples to illustrate aspects of time and space in the context
of chemical transport dynamics:
(1) The classical example of the brachistochrone (ancient Greek: "shortest time"), or path of
fastest descent, is the curve (path) that would carry an idealized point-like body, starting at
rest and moving along the curve, without friction, under constant gravity, to a given end
point in the shortest time. (Correct solution: Johann Bernoulli, 1697) Somewhat non-
intuitively, the path that leads to the shortest travel time is not a straight line, but, rather, a
special curve that is longer than a straight line (a cycloid)! (See:
http://old.nationalcurvebank.org//brach/brach.htm)
(2) What error can be introduced when "averaging" in terms of "space"? Consider the case of
driving a total distance of 100 km, by first traveling 50 km at 1 km h$^{-1}$, and then traveling
50 km at 99 km h$^{-1}$. If we average the speed in terms of space (distance), then we traveled
two segments of 50 km at two speeds, so the average speed is (1 + 99) / 2 = 50 km h$^{-1}$. In
this framework, the total time to travel the 100 km "should" only have been 2 h. However,
in terms of *time*, the travel time is actually 50.5 h.
These simple examples help to emphasize the errors introduced by traditional conceptual
thinking, wherein the effects of *spatial* transport and domain heterogeneity are quantified only on
the basis of spatial characteristics. It is worth recalling, too, Einstein's quantitative treatment of
Brownian motion (Einstein, 1905). Prior to his analysis, researchers applied – with puzzlement –
a time-dependent velocity, *v*, to quantify experimental measurements. Einstein (1905) instead
examined a recursion relation and expansion that led to a diffusion equation whose solution
showed, for the first time, that the root mean squared displacement of particles undergoing
Brownian motion is proportional to $\sqrt{t}$, and not to *vt* as had been assumed traditionally. An
astounding conceptual breakthrough over a century ago, this nature of diffusive motion is now
"common knowledge".


In this same framework of focusing on *time*, the examples shown in Fig. 4 and Fig. 5 emphasize
that for chemical transport, we must recognize the critical role of "*rare events*". These rare events
involve chemical species (migrating "particles" or "packets") that are held up or retained in (while
traveling through, or in/out of) lower velocity regions (pore scale or continuum scale) in the porous
domain, over various periods of time. Such events can have a dominant impact on overall transport
patterns. In this context, one must exercise caution with simple averaging of "small velocity
fluctuations" and effects of molecular diffusion. Rather, small-scale heterogeneities (in both space
and time) do not necessarily "average out" or become insignificant at larger scales; rather, the
effects of "rare events" (e.g., temporary trapping of even small amounts of chemical species via
diffusion into and out of low velocity regions) and fluctuations can propagate and become
magnified, *within and across* length scales from pore to aquifer.
Armed with these thoughts, we suggest that modeling chemical transport has been contentious
for at least three reasons:
1. The desire to work with spatial averaging approaches and equations: The research
community was (and still is) split over the need to recognize and incorporate, *explicitly*,
influences of temporal mechanisms caused largely by spatial heterogeneity (as
characterized by the domain hydraulic conductivity), when formulating "effective" (or
"averaged") equations. And even when recognized, debate remains as to appropriate
mathematical formulation.
2. The lack of data: At least part of the difficulty in developing appropriate models is the lack
of availability of high-resolution laboratory data and field measurements against which
chemical transport models can be tested. Indeed, many elaborate theoretical developments
have been advanced over the decades, with accompanying, analytical and numerical
solutions — and yet, remarkably, comparative studies against actual laboratory data remain
limited, and tests with field measurements are even sparser (see also Sect. 4 for further
discussion of this point).
3. The choice of approach to, and purpose of, chemical transport modeling: Two overarching
approaches to quantifying chemical transport can be defined, focusing on (i) quantification
of "effective", "overall" chemical transport behavior without requiring high-resolution
discretization and numerical solution of the domain, and, alternatively, (ii) high-resolution
hydrogeological delineation and then intensive numerical simulation on highly discretized
grids. We address approaches (i) and (ii) individually, below, in the context also of points
(1) and (2).
The debate in the literature between "effective" and high-resolution hydrogeological modeling,
as well as various preconceptions and misconceptions discussed below and in Sect. 4, lead
naturally to consideration of the (often incorrectly invoked) argument that "fewer model
parameters is better".
We first discuss briefly aspects of high-resolution hydrogeological modeling in Sect. 3.3.1,
and then focus on "effective" transport equation modeling in Sect. 3.3.2. We emphasize that the
latter approach is applicable to both small- and large-scale domains. The former approach is





generally intended for large- (field-)scale systems (although, in a sense, the same approach is often
applied for detailed pore-scale modeling); this approach is not particularly contentious, per se, but
is hampered by the complexity and cost associated with the demand for highly detailed
hydrogeological information. Therefore, research work remains heavily invested in "effective"
transport equation modeling.
**3.3.1 High-resolution domain delineation and modeling**
Efforts to resolve large-scale aquifer systems, to delineate the hydraulic conductivity distribution
at increasingly higher resolutions, began in earnest in the 1990s. Analysis of field sites emphasized
(relatively) high-resolution discretization of domain structure (e.g., "blocks" of the order of 10 m$^3$
at the field scale (Eggleston and Rojstaczer, 1998); $200 \times 200 \times 1$ m$^3$ at large regional scales
(Maples et al. 2019). These efforts, first focusing on determining the fluid flow field, and
subsequently on delineating pathways for chemical transport, began largely because of
dissatisfaction with results of application of 1d, 2d, 3d forms of an "effective" (averaged) ADE
(see further discussion in Sect. 3.3.2). Acquiring high-resolution measurements of structural (e.g.,
mineralogy, porosity) and hydrological properties (e.g., hydraulic conductivity) was made more
feasible in recent years by advances in hydrogeophysics, and as well as by advances in
computational capabilities that enable incorporation of this information in finely-discretized
meshes, and numerical solution for fluid flow and chemical transport.
In these highly resolved (high-resolution) gridded domains, the flow field can be determined
from solution of Darcy's law. Chemical transport is then simulated either by use of streamline
particle tracking methods (accounting for advection and diffusion in a Lagrangian framework) or
via solution of a local (mesh element) continuum-scale transport equation. For chemical transport,
use of an advection-*diffusion* equation might appear preferable – given that it requires no estimate
for the local dispersivity, but some researchers apply an advection-*dispersion* equation, which
necessitates use of mesh-scale dispersivity values that are either assumed or estimated from local
measurements. The latter case assumes mesh-scale transport to be fully Fickian (recall Sect. 3.2)
to quantify overall transport. More recently, we note that alternative formulations of a governing
transport equation that incorporate temporal effects more broadly can also be used in this type of
modeling approach; see, e.g., Hansen and Berkowitz (2020) for incorporation of a continuous time
random walk method (discussed in Sect. 3.3.2) into this modeling framework.
[Parenthetically, we note that "analogous", high-resolution measurements are made at the pore-
scale – in mm to decimeter rock core samples – as a basis for computationally-intensive modeling
of fluid flow and chemical transport at these scales. Similar to the evolution of this approach for
field-scale studies, high-resolution measurements advanced from use of 2d rock micrographs to
advanced micro-computed tomography protocols (e.g., Thovert and Adler, 2011; Bijeljic et al.,
2013; recall Sect. 2.1).]
This approach is attractive in terms of the ability to "reproduce" detailed heterogeneous





hydraulic conductivity structures, and can provide useful "overall assessments" of fluid flow and
chemical transport pathways, and migration of a chemical plume. Moreover, solutions for fluid
flow and chemical transport can be considered "exact", at least at the scale at which the domain is
discretized (and they can thus also capture at least some aspects of non-Fickian transport). But
even at this type of spatial resolution, the ability to effectively quantify actual chemical transport,
even relative to the limited available field measurements, remains a question of debate, and the
research community, as well as practicing engineers, still often prefer to analyze chemical transport
in a domain by use of relatively simple (often 1d, section-averaged) model formulations.

Finally, we point out here that in the context of efforts to obtain increasing amounts of
structural and hydrological information at a given field site, due consideration should also be given
to the "worth" of data. Thus – for example – in an effort to quantify fluid flow or chemical transport
in an aquifer, do we really need "full", detailed knowledge of the system (e.g., porosity, hydraulic
conductivity) at every point in the formation? Possibly non-intuitively, the adage "more data is
better" is often not true, and model incorporation of statistical uncertainty can offer equally
satisfactory solutions with less costly, less measurement- and computationally-intensive, detail
(e.g., Dai et al., 2016).

**653    3.3.2 "Effective" characterization and modeling**


At least since the 1960's, the research community has focused enormous efforts on formulation of
"averaged", or "effective" (often macroscopically, section-averaged 1d) transport equations to
quantify chemical transport, without requiring high-resolution discretization and
(computationally-intensive) numerical solution of the domain. The (now "classical") ADE was
advanced as the governing (partial differential) equation; see also further discussion on "effective
scales of interest", in the context of "upscaling" (Sect. 4). Recall that as discussed in Sect. 3.2, the
ADE assumes Fickian transport behavior, in the sense that mechanical dispersion – which is
defined as an average quantity to describe local fluctuations around the average (advective) fluid
velocity – is treated macroscopically by Fick's law. The classical ADE then specifies coefficients
of longitudinal and transverse dispersivity, which by definition are constants.

Solutions of the ADE were compared against conservative tracer experiments in laboratory
columns (generally 10-100 cm) to produce breakthrough curves (concentration vs. time, at a set
outlet distance), but even from the outset, the applicability of the ADE was questioned by some
researchers (e.g., Aronofsky and Heller, 1957; Scheidegger, 1959). Subsequent flow cell
experiments demonstrated, for example, that the dispersivity constants are not actually constant,
and change with length scale – even over a tens of centimeters – to achieve even *approximate* fits
to the measurements (e.g., Silliman and Simpson, 1987). Moreover, solutions of the ADE appear
inadequate when compared to transport in laboratory flow cells with distinct regions of different
hydraulic conductivities (e.g., Maina et al., 2018). In a sense, then, it can be considered somewhat
surprising that this form of the ADE was subsequently assumed to apply, over several decades, in



a rather sweeping fashion for a wide range of hydrogeological scenarios and length scales. Detailed
discussions of these aspects appear in, e.g., Berkowitz et al. (2006, 2016). [Parenthetically, we
stress again here that if one has complete information at the pore-scale, then solution of the Navier-
Stokes and advection-*diffusion* equations within the pore space can capture the true chemical
transport behavior, i.e., purely spatial information is sufficient to describe chemical transport. But
at continuum scales, time and unresolved heterogeneities became critical, and an "averaged"
equation like the ADE with a "macrodispersion" concept is problematic.]
Moving beyond the ADE, and the definitive need for effective transport equations that quantify
non-Fickian (as well as Fickian) transport (recall Figs. 4 and 5), an alternative approach is to
account for the temporal distribution that affects migration of chemical species, in addition to a
spatial distribution, at a broad continuum level, and employ a transport equation in the spirit of a
"general purpose" ADE. This approach necessarily leads to transport behaviors that are more
general than those indicated by a "general ADE" (i.e., in the context of an overall, averaged 1d
transport scenario for example).
To explain this approach, we refer to the continuous time random walk (CTRW) framework,
which is particularly broad and general. Significantly, and conveniently, it should be recognized
that special (or limit) cases of a general CTRW formulation lead to well-known related "subset"
formulations that also quantify various types of non-Fickian transport. These subsets include
mobile-immobile, multirate mass transfer, and time-fractional derivative formulations, as
explained in, e.g., Dentz and Berkowitz (2003) and Berkowitz et al. (2006). Indeed, in spite of
frequent references to these model formulations as being "different", they are closely related, with
clear mathematical correspondence. Each formulation has advantages, depending on the domain,
problem and objectives of model use; but model selection must first be justified physically, and it
is inappropriate, for example, to apply a mobile-immobile (two domain) model to interpret
chemical transport in a "uniform, homogeneous" porous medium when it displays non-Fickian
transport behavior (recall Fig. 4).
Here, we describe only briefly the principle and basic aspects of the CTRW formulation;
detailed explanations and developments are available elsewhere (e.g., Berkowitz et al., 2006).
To introduce "temporal thinking" in the context of non-Fickian transport, we begin by
mentioning the analogy between a classical random walk (RW) – which leads to Fick's law – and
the CTRW. A classical random walk is given in Eq. 1:

$$P_{n+1}(\ell) = \sum_{\ell'} p(\ell, \ell')\, P_n(\ell') \tag{1}$$

where $p(\ell, \ell')$ represents the probability of a random walker ("particle") advancing from location
$\ell'$ to $\ell$, $P_n(\ell')$ denotes the probability of a particle being located at $\ell'$ at (fixed) time step $n$, and
$P_{n+1}(\ell)$ denotes the probability of the particle then being located at $\ell$ at step $n+1$. With this
formulation in mind, Einstein (1905) and Smoluchowski (1906a,b) demonstrated that for $n$
sufficiently large and a sufficient number of particles undergoing purely (statistically) random
movements *in space*, the spatial evolution of the particle distribution is equivalent to the solution


of the (Fickian) diffusion equation. This elegant discovery demonstrated that a partial differential
equation and its solution can be represented by following, numerically, the statistical movement
of particles (i.e., particle tracking) following a random walk. Generalizing the partial differential
equation to include transport by advection, solution of the ADE under various boundary conditions
can then be determined by an appropriate random walk method.
The simple random walk given in Eq. 1 can be generalized by accounting for time, replacing
the particle transition (or iteration) counter $n$ by a time distribution. The generalized formalism in
Eq. 2, with the joint distribution $\psi(\mathbf{s}, t)$, called "continuous time random walk" and applied to
transport, was first introduced by Scher and Lax (1973):
$$R_{n+1}(\mathbf{s}, t) = \sum_{s'} \int_0^t \psi(\mathbf{s} - \mathbf{s}', t - t') R_n(\mathbf{s}', t') dt' \qquad (2)$$

where $R_{n+1}(\mathbf{s}, t)$ is the probability per time for a particle to just arrive at site $\mathbf{s}$ at time $t$ after $n+1$
steps and $\psi(\mathbf{s}, t)$ is the probability rate for a displacement from location $\mathbf{s}'$ to time $\mathbf{s}$ with a difference
of arrival times of $t$-$t'$. It is clear that $\psi(\mathbf{s}, t)$ is the generalization of $p(\ell, \ell')$ in Eq. 1, and that the
particle steps can each now take place at different times. Indeed, it is precisely this explicit
accounting of a distribution of temporal contributions to particle transport, not just spatial
contributions, that offers the ability to effectively quantify transport behaviors as expressed by,
e.g., heavy-tailed, non-Fickian particle arrival times.
To where does the generalization in Eq. 2 lead us? In a mindset similar to that of Brownian
motion, and Einstein's 1905 breakthrough mentioned above at the outset of Sect. 3.3, a puzzle
arose about seven decades later for researchers attempting to interpret observations of electron
transit times in disordered semiconductors. The electron mobility (defined as velocity per unit
electric field), which was considered an intrinsic property of the material, was found to depend on
variables that changed the duration of the experiment, such as sample length or electric field. Scher
and Montroll (1975), considering Eq. 2, discovered that the mean displacement $\overline{\ell}$ of the electron
packet does not advance as $\overline{\ell} = vt$, but rather as $\overline{\ell} \sim t^\beta$.
In the context of chemical transport in geological formations, the behavior $\overline{\ell} \sim t^\beta$ can be
attributed to a wide distribution of transition times in naturally disordered geological media. In the
CTRW formulation, the transition time distribution is characterized by a power law of the form
$\psi(t) \sim t^{-1-\beta}$ for $t \to \infty$ and $0 < \beta < 2$; significantly, the resulting transport behavior is Fickian for
$\beta > 2$. At large times, for this $\psi(t)$ dependence, the mean displacement $\overline{\ell}(t)$ and standard deviation
$\overline{\sigma}(t)$ of the migrating chemical plume $c(\mathbf{s}, t)$ scale as $\overline{\ell}(t) \sim t^\beta$ and $\overline{\sigma}(t) \sim t^\beta$ for $t \to \infty$, $0 < \beta <$
1 (Schlesinger, 1974). Moreover, for $t \to \infty$ with $1 < \beta < 2$, the plume scales as $\overline{\ell}(t) \sim t$ and
$\overline{\sigma}(t) \sim t^{(3-\beta)/2}$. These behaviors are notably different than that of Fickian transport models, for
which (from the central limit theorem), $\overline{\ell}(t) \sim t$ and $\overline{\sigma}(t) \sim t^{1/2}$.
With the concepts described here, and using the generally applicable decoupled form $\psi(\mathbf{s}, t) =$
$p(\mathbf{s})\psi(t)$, where $p(\mathbf{s})$ is the probability distribution of the transition lengths and $\psi(t)$ is the
probability rate for a transition time $t$ between sites, Eq. 2 can be developed into an (integro-)partial



differential equation. Thus, the ADE given by

$\frac{\partial c(\mathbf{s},t)}{\partial t} = -[\mathbf{v}(\mathbf{s}) \cdot \nabla c(\mathbf{s},t) - \mathbf{D}(\mathbf{s}){:}\nabla^2 c(\mathbf{s},t)]$      (3)

where $c(\mathbf{s}, t)$ is the concentration at location $\mathbf{s}$ and time $t$, $\mathbf{v}(\mathbf{s})$ is the velocity field and $\mathbf{D}(\mathbf{s})$ is the
dispersion tensor, is replaced by the more general CTRW transport equation:


$\frac{\partial c(\mathbf{s},t)}{\partial t} = -\int_0^t M(t-t')\left[\mathbf{v}_\psi \cdot \nabla c(\mathbf{s},t') - \mathbf{D}_\psi : \nabla\nabla c(\mathbf{s},t')\right]dt'$      (4)

where $\mathbf{v}_\psi$ and $\mathbf{D}_\psi$ are generalized particle velocity and dispersion, respectively, and $M(t)$ is a
temporal memory function based on $\psi(t)$.

The strength of this type of formulation is that it effectively quantifies (non-Fickian) early

arrivals and late time tailing of migrating chemical species, and the spatial evolution of chemical
plumes in heterogeneous media. For example, recalling the scenario in Fig. 2, wherein 99.9% of
the inflowing particles traverse the preferential pathways seen in Fig. 2c, detailed numerical
simulations indicate that concentration breakthrough curves exhibit significant, non-Fickian, long-
time tails (Edery et al., 2014). Choice of an appropriate power-law form of $\psi(t)$ was then shown
to capture this behavior; moreover, a functional form defining the value of the power-law exponent
$\beta$ in $\psi(t)$ was identified, based on statistics of the hydraulic conductivity and particle interrogation
of the domain (Edery et al., 2014).

Equation (4) is essentially an ADE weighted by a temporal memory. When $\psi(t)$ is an

exponential function (or power law but for $\beta \geq 2$), $M(t) \to \delta(t)$ and we recover Fickian transport
described by the ADE; thus, the ADE assumes, implicitly, that particle transition times are
distributed exponentially. But with a power law form $\psi(t) \sim t^{-1-\beta}$ for $0 < \beta < 2$, the transport is
non-Fickian. A wide range of functional forms of $\psi(t)$ can be chosen, including, e.g., truncated
power law forms that allow evolution to Fickian transport at large times or travel distances (e.g.,
Berkowitz et al., 2006), as well as Pareto (e.g., Hansen and Berkowitz, 2014) and curved (or
inverse gamma; e.g., Nissan and Berkowitz, 2019) temporal distributions. Other, generally
simpler, choices of $\psi(t)$ or $M(t)$ lead to mobile-immobile, multirate mass transfer, and time-
fractional derivative formulations. We note, too, that the elegant result derived by Gelhar and
Axness (1983) and others, discussed in Sect. 1.2, is valid only at an asymptotic limit, wherein
transport is Fickian and there is no residual non-Fickian memory in the plume advance.

Each of these power law forms of course requires one or more parameters – at least $\beta$ – and in

some cases, other parameters that define, e.g., a transition time from non-Fickian to Fickian
transport (Berkowitz et al., 2006; Hansen and Berkowitz, 2014; Nissan and Berkowitz, 2017).
These parameters have physical meaning, and are not purely empirical; perspectives on "numbers
of parameters" associated with all models are discussed in Sect. 3.3.3. The question of how model
parameter values are determined is addressed in Sect. 4.1.

The efficacy of formulations that incorporate, whether explicitly or implicitly, some type of


power-law characterization of temporal aspects of chemical transport, is now generally recognized
in the literature. Indeed, applications of mobile-immobile, multirate mass transfer, time-fractional
advection-dispersion, and general CTRW formulations have been applied quite extensively and
successfully. In particular, solutions of Eq. 4 and related variants have interpreted a wide range of
chemical transport scenarios: (i) pore-scale to meter scale laboratory experiments, field studies,
and numerical simulations, in (ii) porous, fractured, and fractured porous domains, (iii) accounting
for constant and time-dependent velocity fields, and (iv) for both conservative and reactive
chemical transport scenarios. Solutions to address some of these scenarios are more easily obtained
by use of particle tracking methods that incorporate the same considerations and power-law form
of $\psi(t)$, as embedded in Eq. 4.
It should be recognized that, like the ADE, Eq. 3, the formulation given in Eq. 4 represents a
continuum-level mechanistic model. Discussion in the literature about the need for "mechanistic
models" often uses the term rather loosely: "mechanistic" transport model equations are based on
fundamental laws of physics, with (constant) parameters that have physical meaning (e.g.,
hydraulic conductivity, diffusivity, sorption), and thus offer process understanding. But to quantify
the spatiotemporal evolution of a migrating chemical plume, additional parameters are needed. We
argue that, because of the nature of geological materials, a transport equation should of course
capture the relevant physical (and chemical, if the species is reactive) mechanisms that impact the
transport, but to do so, we must *also* capture the uncertain characterization of hydrogeological
properties dues to the reality of unresolved (unmeasurable) heterogeneities at any length scale of
interest. Thus, we suggest that a *mechanistic-stochastic equation formulation* such as given in Eq.
4 (which incorporates a probability density function to account for temporal transitions that cannot
be determined only from spatial information) is required, describing known transport mechanisms
(and with physically meaningful parameters), and accounting for unknown (and unknowable!)
information.
We note here, too, that other stochastic continuum averaging methods have been proposed in
the literature, in the same context of efforts to formulate a "general", "effective" transport equation
at a specific scale of interest (see further discussion on "effective" equations and "upscaling" in
Sect. 4). In many cases, though, sophisticated stochastic averaging and homogenization
approaches have led to transport formulations that are essentially intractable, in terms of solution,
and/or have remained at the level of hypothesis without being tested successfully against actual
data.

**3.3.3 Are fewer parameters always better? (Answer: No!)**

The term "modeling" is used in many contexts and with differing intents. However, in the
literature dealing with chemical transport in subsurface hydrological systems, there are frequent
(often misguided!) "arguments" regarding "which model is better", with a major point of some
authors being the claim that "fewer parameters is always best". Not always. Indeed, some models





involve more parameters than others, but if these parameters have physical meaning and are needed
as factors to quantify key mechanisms, then "more parameters" is not a "weakness". We
emphasize, too that when weighing use of a specific (any!) model, "better" also depends (at least
in part) on what the modeling effort is addressing; clearly – and regardless of the number of
parameters – a "back-of-the-envelope" calculation using a simple model is sufficient if, for
example, one requires only an order of magnitude estimate of the center-of-mass velocity of a
migrating contaminant plume (or in other words, no need for artillery to swat a mosquito!) [In this
context, quoting Albert Einstein regarding his simplification of physics into general relativity:
"Everything should be made as simple as possible, but not simpler."]
Considering chemical transport in subsurface geological formations, and the aim of
quantifying (modeling) the evolution of a migrating chemical plume in both space and time, we
return to focus on the ADE- and CTRW-based formulations discussed in Sect. 3.3.2. As noted in
the preceding sections, CTRW formulations have been demonstrated repeatedly to describe a
system effectively, with sufficient parameters to account for the various mechanisms. Most
significantly, the seminal work of Scher and Montroll (1975) showed that the $\beta$ exponent *had* to
be included because the mean displacement was not linear with time (i.e., the mean displacement
$\overline{\ell}$ of the electron packet does not advance as $\overline{\ell} = vt$, but rather as $\overline{\ell} \sim t^{\beta}$ ). Thus, a corresponding
parameter, relative to an ADE formulation invoking Fickian transport, is unavoidable in such
cases. In a sense, too, one can argue that this parameter is not an "additional" parameter relative
to the ADE; rather, a decision to invoke the ADE to quantify a transport problem simply assigns a
value to this parameter, implicitly, as $\beta \geq 2$.
It should be recognized that – while not (yet?) standard practice in the research community –
quantitative model information criteria, or model selection criteria, can be used to assess and
compare various model formulations that are applied to diverse scenarios (such as fluid flow,
chemical transport) in subsurface geological formations. These information criteria include AIC
(Akaike, 1974), AICc (Hurvich and Tsai, 1989), and KIC (Kashyap, 1982) measures, as well as
the Bayesian (or Schwarz) BIC (Schwarz, 1978). They are formulated to rank models, or assign
(probabilistic) posterior weights to various models in a multimodel comparative framework, and
therefore focus on model parameter estimates and the associated estimation uncertainty. As such,
these information criteria discriminate among various models according to (i) the ability to
reproduce system behavior, and (ii) the structural complexity and number of parameters.
Discussion of theoretical and applied features of these criteria is given elsewhere (e.g., Ye et al.,
866  2008).

Specifically in the context of the ADE and CTRW formulations, with an accounting also of
chemical reactions, for example, it was shown that while solution of an ADE can fit measurements
from some locations quite closely, the CTRW formulation offers significantly improved predictive
capabilities (in the context of model assessment in the presence of uncertainty) when examined
against an entire experimental data set (Ciriello et al., 2015). In addition, focusing on the most
sensitive observations associated with the CTRW model provides a stronger basis for model
prediction, relative to the most sensitive observations corresponding to the ADE model.



To conclude this section: Notwithstanding the above arguments, some readers might continue
to argue that the approach discussed here – viz., the need for time considerations as well as space
(as embodied in the CTRW framework and related formulations) – is "inelegant" because it
requires more parameters relative to the classical ADE. In response, the reader is encouraged to
recall the words of Albert Einstein following criticism that his theory of gravitation was "far more
complex" than Newton's. His response was simply: "If you are out to describe the truth, leave
elegance to the tailor".
**4 The holy grail of upscaling, and myths about "a priori" parameter determination**
We begin by defining the term "upscaling" in the context of the discussion here on chemical
transport. As defined in the Introduction, Sect. 1.2, we use the term "upscaling" to describe the
effort to develop and apply chemical transport equations at large length scales, and identify
corresponding model parameter values, based on measurements and parameter values obtained at
significantly smaller length scales.
We attempt "upscaling" in the hope of developing governing equations for chemical transport
at larger and larger scales, from pore, to core, to plot, and to field length scales. Clearly, then,
"upscaling" is relevant to the modeling approach discussed in Sect. 3.3.2 – which focuses on use
of "averaged", or "effective" (often 1d, or section-averaged) transport equations – and not to the
high-resolution domain delineation and modeling approach of Sect. 3.3.1.
However, in light of the discussion in Sect. 2 and Sect. 3, we argue that "upscaling" of chemical
transport equations is very much a holy grail. Particularly in light of recognizing temporal effects,
in addition to spatial characterization, we maintain that it is necessary to formulate, calibrate and
apply models using measurements at similar scales of interest, in both space and time. Of course,
similar equation *formulations* can be applied at different spatial scales. But parameter values for
transport equations cannot generally be determined a priori or from purely spatial or flow-only
measurements; *measurements with a temporal "component", at the relevant length scale of*
*interest, are required.*
In Sect. 4.1, we briefly discuss aspects of model calibration. This leads naturally to our
discussion of upscaling in Sect. 4.2.
**4.1 Parameter determination and model calibration**
First, it is prudent to offer some words about the need for parameter estimation, or model
calibration. Unless one is dealing with first principles calculations of a physical process (e.g.,
molecular diffusion) in a perfectly homogeneous domain, a priori determination of model
parameters – for *any* model equation formulation – requires calibration against actual experimental
measurements; in some limited cases, detailed numerical simulations can be used at small (pore)


scales (e.g., using an advection-*diffusion* equation with the fluid phase, together with solution of
the Navier-Stokes equations to first determine the precise flow field in the pore space). Indeed,
then, at any realistic problem or scale of interest, *all* chemical transport models require calibration.
This fundamental tenet should be clear and well-recognized, yet the literature contains all-too-
frequent – and both misguided and misleading – "criticism" of various model formulations,
claiming that "parameters are empirical because they are estimated by calibration (fitting) to
experiments"; additional "criticisms" follow, for example, that such as a model is therefore not
"universal", and/or "it therefore has no predictive capability". We address these latter "criticisms"
in Sect. 4.2. Parameters are *not* "empirical" simply because their values are determined by
matching to an experiment! Moreover, it should be recognized that application even of the classical
ADE at various column and larger scales requires estimates – obtained by calibration – of
dispersivity coefficients (and for high-resolution domain delineation and modeling as discussed in
Sect. 3.3.1, "block-scale" dispersivities are needed). [Note: And if dispersivities are not actually
determined for a specific experiment, but selected from on the literature for "typical" values of
dispersivity, there is still a reliance on calibration from previous "similar" studies!] Moreover, with
reference to the desire for model parameters that represent fundamental, *spatial* hydrogeological
properties of the domain, note that even the classical ADE dispersivity parameter is not uniquely
identified with such properties; rather, it varies even in a given domain as a function of chemical
plume travel distance or time.
With regard to model "universality", recall that, for example, percolation theory (discussed at
the beginning of Sect. 2) offers "universal" exponents in scaling relationships. But even for this
type of convenient and useful, statistical model, such scaling relationships, too, can only advance
from "scaling" (e.g., $A \sim B$) to a full "equation" (e.g., $A = kB$) by calibration of a coefficient of
equality ($k$) against actual measurements. So even in "simple" models, model calibration cannot
be avoided.
To address "empiricism" – here enters the question of whether parameters of a particular model
(in this case, equations for chemical transport) have a physical meaning. As discussed in Sect.
3.3.2, a *mechanistic-stochastic equation formulation* such as given in Eq. 4 incorporates a
probability density function to describe known transport mechanisms in a stochastic sense; but
stochastic does not mean "unphysical", and the parameters as given in, e.g., particular functional
forms of $M(t)$ or $\psi(t)$ are indeed physically meaningful. For example, the key $\beta$ exponent
characterizing the power law behavior can be linked directly to the statistics of the hydraulic
conductivity field (Edery et al., 2014), or, in a fracture network, be determined from the velocity
distribution in fracture segments (Berkowitz and Scher, 1998), which is related directly to physical
properties of the domain. Similarly, corresponding parameters appearing in "subset" formulations
to quantify non-Fickian transport – e.g., mobile-immobile, multirate mass transfer, and time-
fractional derivative formulations – can be understood to have physical meaning (e.g., Dentz and
Berkowitz, 2003; Berkowitz et al., 2006). These parameters, too, of course require determination
by model calibration to experimental data (or where appropriate, to results of numerical
simulations), just as for any other model, including ADE formulations.



**4.2 Upscaling, the scale of interest, and predictive capabilities**

Upscaling of *fluid flow* "works" because pressure propagation is essentially instantaneous. At the Darcy scale – which is the "practical" scale for most applications – flow paths and streamlines do not change with increasing gradient (as long as a transition to turbulent flow is not reached), the equation formulation remains valid, and the fluid residence time in a domain is irrelevant because self-diffusion of water does not affect overall fluid fluxes. Pore-scale flow analyses are local and more specialized, and "upscaling" is not per se an objective.

For chemical transport, though, the situation is totally different. Why? Because "upscaling" entails some kind of "coupled" averaging in both space *and* time, and it is far from clear how, if at all, this can be achieved. Moreover, small-scale concentration fluctuations do not necessarily "average out", but instead propagate from local to larger spatial scales. To illustrate another aspect of the complexity, the Péclet number (Pe) in heterogeneous media, with preferential pathways, varies locally in space (recall Fig. 3 and the discussion in Sect. 3.1). Averaging to obtain a macroscale ("upscaled") Pe must address the relative, locally varying effects of advection and diffusion in space, as well as the overall residence time in the domain; after all, it is these effects that dominate determination of the plume evolution. Thus, upscaling requires spatial averaging, but (at least an) *implicit* temporal averaging must also be included. It can be argued that no single, effective Pe can be defined for the entire domain; whether or not it is possible, and how, it is possible to average local Pe values to achieve a single, meaningful domain-scale Pe remains an open question. And whether we like it or not, even with complete information on the spatial (local) Pe distribution, the impact on the overall transport pattern evolution cannot be determined without actually solving for transport in the domain.

For chemically reactive species, the transport situation becomes even more complex, because the local residence time, not just the local Pe, must be taken into consideration. Moreover, when precipitation or dissolution processes are present, the velocity field will change locally, introducing additional local temporal and spatial variability. And when sorption is present but tapers off (when the cation exchange capacity is met, for example), even the diffusion coefficient itself changes. These factors further complicate attempts to upscale. [In this context, too, it should be noted that for chemically-reactive systems, it is well-known (e.g., White and Brantley, 2003) that there is often a significant lack of correspondence between laboratory and field-based estimates of geochemical reaction rates and rates of rock weathering, with field-scale estimates – often based on macroscopically Fickian, ADE-like transport formulations – being generally significantly smaller.]

Thus, we suggest that focusing efforts on attempting to develop upscaling methodologies for chemical transport, based on any transport equation formulation, appear to be doomed largely to failure – as evidenced, too, by decades of research publications. Rather, we argue that because of the subtle effects of temporal mechanisms, and their close coupling to spatial mechanisms, use of



an "effective", or "averaged" continuum-level equation to describe chemical transport (as opposed, e.g., to intensive numerical simulation using a streamline particle tracking method in a high-resolution hydraulic conductivity field) requires calibration of a suitable model at the appropriate scale of interest, with model parameter values calibrated at essentially the same scale.

We emphasize, though, that as stated at the outset of Sect. 4, we do argue that similar (continuum-level) transport equation *formulations* can be applied at different spatial scales, as long as they are mechanistically correct (with a *temporal* component), and the parameter values are based on *measurements at the relevant length scale of interest.*

Now, in the context of the above arguments regarding "upscaling" and model application, we return to the ideas presented in Sect. 3.3.2 and consideration of model formulations that account for both spatial and temporal effects. We first mention use of the ADE. As pointed out in Sect. 3.2 and extensive literature, the "constant" (as required by the ADE formulation) "intrinsic" dispersivity parameter changes significantly even over relatively small (e.g., 10's of cm's, Silliman and Simpson, 1987) increases in length – and therefore also time – scales, so that it makes no real sense to attempt to define an "upscaled" dispersivity parameter for larger scales. Even in the framework of high-resolution domain delineation and modeling, discussed in Sect. 3.3.1 – which is not "upscaling" as defined here – the question remains as to what dispersivity values are relevant for field-scale aquifer "blocks" of the order of 100 to 1000's of m$^3$.

In contrast, CTRW and related transport formulations with explicit accounting of time effects, as outlined in Sect. 3.3.2, can be applied meaningfully to interpret real measurements and transport behavior at "all" scales. We *can* use the same equation formulation at different scales, with different but relevant parameters at each scale. We emphasize, too, that we do not argue for "hard" length scales, so that in principle, e.g., an appropriate (CTRW-based) model calibrated at 20 cm will be applicable to 100 cm scales, and that a model calibrated on a 100 m scale data set can be applicable at a kilometer scale. The point, though, is that it makes no sense to calibrate at a centimeter scale and then expect to somehow "upscale" parameters to apply the same model at a kilometer scale. [Note: As an aside, over very large field-length and field-time scales, we point out that homogenization effects of molecular diffusion *may* become more significant, lessening impacts of some preferential pathways.] Similarly, a CTRW-based approach can be applied over a range of *time* scales, because the power law accounting for temporal effects can be as broad as needed. In these cases, temporal effects are critical, because at the continuum (Darcy) scale, streamlines do not change but residence times do. Specifically, for example, a model formulation with a fixed set of parameters can interpret transport measurements in the same domain, but acquired under different hydraulic gradients (fluid velocities), and thus domain residence times (Berkowitz and Scher, 2009). Indeed, because of the temporal accounting, CTRW has been applied successfully over scales from pores (e.g., Bijeljic et al., 2013) to kilometers (e.g., Goeppert et al., 2020), with parameter calibration at the relevant scale of interest. In principle then, too, a calibrated model shown to be effective/meaningful over one region of a porous medium or geological formation can offer at least a reasonable estimate of transport behavior elsewhere in the medium/formation, at a similar length/time scale, and as long as the medium/formation can be



expected to have reasonably similar hydrogeological structure and properties.
And finally, another critical aspect must be pointed out with regard to *continuum-scale*
transport models as outlined in Sect. 3.2. The preceding discussion leads to the stated need and
desire – at least in principle – to achieve model "prediction". This term appears often, but it is
often used incorrectly. Fitting a model solution to data is of course not "prediction". On one hand,
using specific experiments and data sets, models can be used to *characterize* transport behavior,
e.g., is transport Fickian or non-Fickian?, or, is a migrating chemical plume compact or elongated
and ramified?, which is of fundamental importance. But if *prediction* is the ultimate goal
(recognizing that addressing *prediction uncertainty* is yet another consideration), then we require
multiple data sets from the same porous medium or geological formation, in the sense that we need
measurements over a range of length scales, and/or over a range of time scales (i.e., same distance,
different flow rates). An intended model can then be calibrated against one part of the data set; the
calibrated model is then applied "as is" and the resulting solution ("prediction") is compared
against other ("previously unknown") measurements. At the laboratory scale, such a protocol is
feasible, but rarely executed. Rather, the literature generally reports fits of transport equation
solutions *at specific scales* (individual data sets at a given length scale), and not over a range of
scales, so that no real testing of "upscaling" or "prediction" is achieved. Thus, even at laboratory
scales, true "predictive capability" of a model is rarely examined or reported. [Note: A similar
approach to "prediction" can be done in a purely numerical/computational study, using "ground
truth simulations" that are *assumed* correct (e.g., Darcy flow calculations and then streamline
particle tracking for chemical transport in a highly-resolved domain; recall Sect. 3.3.1), and then
comparing solutions from a continuum (e.g., partial differential equation) model solution. But it
should be recognized that results from *assumed* simulation methods are often unsatisfactory when
compared against experimental measurements and field observations.]
And at the field scale, the situation is even less satisfying; large-scale field tests for chemical
transport are difficult and expensive to execute, so that systematic data sets that enable testing of
model "prediction" are essentially non-existent. Moreover, at the field scale, there are necessarily
highly limited numbers of measurements, so that oft-used (and non-unique) interpolation of sparse
concentration measurements employed to yield (ideally 3d) contour maps of concentration will
unrealistically smooth and dampen non-uniform, ramified and irregular preferential pathways
(recall Fig. 2). Thus, notwithstanding the extensive research efforts reported in the literature, truly
comparative studies using field measurements – to genuinely test proposed "upscaling"
methodologies – are essentially non-existent (e.g., Berkowitz et al., 2016)! In this context, then,
we note that criticism in the literature that a given continuum model "demonstrates no predictive
capability" is in fact not generally based on its assessment relative to sufficiently resolved,
representative, and real data sets.
Finally, it is important to recognize that models are most commonly tested against 1d, section-
averaged concentration breakthrough curves, which can be (i) measured directly in laboratory
column experiments, (ii) estimated or derived in 2d/3d laboratory flow cells by averaging over
control planes, or (iii) estimated from limited monitoring well measurements (single or multi-level



sampling with depth) at a fixed number of locations. The latter case, in particular, requires extensive interpolation and/or assumption of a large-scale, essentially 1d and uniform, macroscopic flow field. Moreover, chemical transport model discrimination often requires breakthrough curves that extend over the late time tailing, which are particularly difficult to determine in field conditions, due both to interruptions or lack of practicality in well monitoring at long times, and to detection limits of measurement methods. While reliance on such 1d (section averaged, over some control plane) breakthrough curves many not be ideal, it is *reality* in terms of feasible data acquisition. As a direct consequence, model selection, model parameter fits, and model calibration results may each (and all) be non-unique and lead to confusing or conflicting conclusions. It is therefore critical that we at least select from mechanistic-stochastic models based on fundamental laws of physics, with parameters that have physical meaning, as discussed in Sect. 3.3.2, rather than from models invoking purely statistical distributions or assumptions known to be incorrect.

**5 Concluding remarks**

The ideas, arguments and perspectives offered here represent an effort to somehow summarize and synthesize understanding of existing approaches and methods proposed to quantify chemical transport in subsurface hydrological systems. The literature on this subject is vast, extending over decades, and measurements and observations of chemical transport range from pore-scale microfluidic laboratory cells to aquifer field scales. A similarly broad range of model formulations has been proposed to quantify and interpret these measurements/observations. And yet, outcomes of these efforts are often largely unsatisfactory.

We contend that modeling obstacles arise largely because domain heterogeneity – in terms of porosity, hydraulic conductivity, and geochemical properties – is present over multiple length scales, so that "unresolved", practically unmeasurable heterogeneities and preferential pathways arise at every length scale. Moreover, while *spatial* averaging approaches are effective when considering overall fluid flow – wherein pressure propagation is essentially instantaneous and the system is "well mixed" – purely *spatial* averaging approaches are far less effective for chemical transport, essentially because well-mixed conditions do not prevail. We assert here that an explicit accounting of *temporal* information, under uncertainty, is an additional – but *fundamental* – component in an effective modeling formulation. As a consequence, we argue that for continuum-scale analysis, mechanistic-stochastic models such as those outlined in Sect. 3.2 must be invoked to account explicitly for both "additional" temporal effects and unresolved heterogeneity. Clearly, no single model is "best" for all situations and objectives, but any selected model must be physically relevant and justified.

We further assert, as an outcome of these arguments, that "upscaling" of chemical transport equations – in the sense of attempting to develop and apply chemical transport equations at large (length) scales based on measurements and model parameter values obtained at significantly



smaller length scales – is very much a holy grail. Rather, because probabilistic considerations required to account for small-scale fluctuations do not necessarily "average out" (and can propagate from local to larger spatial scales), we maintain that it is necessary to formulate, calibrate and apply models using measurements at similar scales of interest, in both space and time.

In all of our efforts to reasonably model chemical transport in subsurface hydrological systems, we should recognize and accept the objective of advancing our science by integrating theory, computational techniques, laboratory experiments and field measurement, with the aim of extracting broadly applicable insights and establishing practical, functional tools. In this context, as a close colleague and mentor said to me many, many years ago, "remember, this is hydrology, with very real problems to address…we're not doing string theory"!

We have included many points for discussion and open thought. The reader may not agree with all arguments and conclusions raised here, but scholarly debate is critical: it is hoped that this contribution will stimulate further discussion, assist in ordering classification of the (often confusing) terminologies and considerations, and identify the most relevant, real questions for analysis, implementation and future research.

We hope that the above thoughts and illustrations (i) encourage careful consideration prior to data collection (whether from field measurements, laboratory experiments, and/or numerical simulations), (ii) assist in experimental design and subsequent analysis, and, even more significantly, (iii) influence the research agenda for the field by challenging researchers to ask and address appropriately formulated questions. In terms of "modeling" efforts: recall the statement by Manfred Eigen (Nobel prize chemistry, 1967): "A theory has only the alternative of being right or wrong. A model has a third possibility: it may be right, but irrelevant."

*Data availability.* All data have been reported and published previously, as given in the relevant citations.

*Author contributions.* Single author contribution.

*Competing interests.* The author is a member of the editorial board of Hydrology and Earth System Sciences. The peer-review process was guided by an independent editor, and the author has no other competing interests to declare.

*Acknowledgements.* This contribution has emanated from research and discussions – often animated and always thought-provoking – with students, post-doctoral fellows, and more senior colleagues in my group and from around the world, over the last two decades. I am deeply grateful to them all! I also appreciate critical insights and constructive comments by Alberto Guadagnini and Harvey Scher. B.B. holds the Sam Zuckerberg Professorial Chair in Hydrology.

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
