# Peer review of "HESS Opinions: Chemical transport modeling in subsurface hydrological systems – Space, time, and the "holy grail" of "upscaling""

_Hydrology and Earth System Sciences, 2021_

## Referee Comment (RC2)

**HESS Opinions: Chemical transport modeling in subsurface hydrological**
**systems – Space, time, and the holy grail of "upscaling"**

*[handwritten: IN CONTEMPORARY PARLANCE HOLY GRAIL IS SOMETIMES USED TO REFER TO AN ATTAINABLE OBJECTIVE OR WISH. NOT AN IDEAL CHOICE OF TERMS.]*

Brian Berkowitz[1]
[1]Department of Earth and Planetary Sciences, Weizmann Institute of Science, Rehovot 7610001,
Israel
*Correspondence to:* Brian Berkowitz (brian.berkowitz@weizmann.ac.il)

**Abstract.**

Extensive efforts over decades have focused on quantifying chemical transport in subsurface
geological formations, from microfluidic laboratory cells to aquifer field scales. Outcomes of these
efforts have remained largely unsatisfactory, however, largely because domain heterogeneity (in
terms of, e.g., porosity, hydraulic conductivity, geochemical properties) is present over multiple
length scales, and "unresolved", practically unmeasurable heterogeneities and preferential
pathways arise at virtually every scale. While *spatial* averaging approaches are effective when
considering overall fluid flow – wherein pressure propagation is essentially instantaneous and the
system is "well mixed" – purely *spatial* averaging approaches are far less effective for chemical
transport, essentially because well-mixed conditions do not prevail. We assert here that an explicit
accounting of *temporal* information, under uncertainty, is an additional, but *fundamental*,
component in an effective modeling formulation. As an outcome, we further assert that "upscaling"
of chemical transport equations – in the sense of attempting to develop and apply chemical
transport equations at large (length) scales, based on measurements and model parameter values
obtained at significantly smaller length scales – is very much a holy grail. Rather, we maintain that
it is necessary to formulate, calibrate and apply models using measurements at similar scales of
interest, in both space and time.

**Keywords:** Preferential flow, anomalous transport, numerical modeling, measurements

*[handwritten annotations: "MEANING HERE NOT CLEAR: THEY IS NOT JUSTIFIED"; "MANY OF US DID NOT EXPECT MORE THAN WAS FOUND, SO ARE NOT UNSATISFIED, BUT RATHER APPRECIATING NATURE'S COMPLEXITY."; "DOES THIS MEAN A MYTHICAL AND UNOBTAINABLE GOAL, OR A VALID OBJECTIVE? I GUESS THE FORMER."; "KEY POINT: IF EVERY MODEL NEEDS TO BE CALIBRATED @ ALL SCALES, IS THE MODELING EXERCISE FUTILE?"]*

[revised manuscript text omitted]

*[handwritten: YOU INTRODUCE AIC, BUT DON'T PROVIDE ANY DATA ON HOW THE CTRW MEASURES UP BY THIS METRIC. ALSO, YOU DON'T PRESENT DATA ON PARAMETER INDEPENDENCE, OR STANDARD ERROR.]*

To conclude this section: Notwithstanding the above arguments, some readers might continue
to argue that the approach discussed here – viz., the need for time considerations as well as space
(as embodied in the CTRW framework and related formulations) – is "inelegant" because it
requires more parameters relative to the classical ADE. In response, the reader is encouraged to
recall the words of Albert Einstein following criticism that his theory of gravitation was "far more
complex" than Newton's. His response was simply: "If you are out to describe the truth, leave
elegance to the tailor".

**4 The holy grail of upscaling, and myths about "a priori" parameter determination**

We begin by defining the term "upscaling" in the context of the discussion here on chemical
transport. As defined in the Introduction, Sect. 1.2, we use the term "upscaling" to describe the
effort to develop and apply chemical transport equations at large length scales, and identify
corresponding model parameter values, based on measurements and parameter values obtained at
significantly smaller length scales.

We attempt "upscaling" in the hope of developing governing equations for chemical transport
at larger and larger scales, from pore, to core, to plot, and to field length scales. Clearly, then,
"upscaling" is relevant to the modeling approach discussed in Sect. 3.3.2 – which focuses on use
of "averaged", or "effective" (often 1d, or section-averaged) transport equations – and not to the
high-resolution domain delineation and modeling approach of Sect. 3.3.1.

However, in light of the discussion in Sect. 2 and Sect. 3, we argue that "upscaling" of chemical
transport equations is very much a holy grail. Particularly in light of recognizing temporal effects,
in addition to spatial characterization, we maintain that it is necessary to formulate, calibrate and
apply models using measurements at similar scales of interest, in both space and time. Of course,
similar equation *formulations* can be applied at different spatial scales. But parameter values for
transport equations cannot generally be determined a priori or from purely spatial or flow-only
measurements; *measurements with a temporal "component", at the relevant length scale of*
*interest, are required.*

*[handwritten left margin: YOU MEAN UNSUSTAINABLE?]*
*[handwritten right margin: PROOF?]*

In Sect. 4.1, we briefly discuss aspects of model calibration. This leads naturally to our
discussion of upscaling in Sect. 4.2.

**4.1 Parameter determination and model calibration**

First, it is prudent to offer some words about the need for parameter estimation, or model
calibration. Unless one is dealing with first principles calculations of a physical process (e.g.,
molecular diffusion) in a perfectly homogeneous domain, a priori determination of model
parameters – for *any* model equation formulation – requires calibration against actual experimental
measurements; in some limited cases, detailed numerical simulations can be used at small (pore)

*[handwritten right margin: WHAT ABOUT WITH CT SCAN? WOULD WE NOT BE ABLE, SOME TO ESTIMATION BASED ON GEOMETRY?]*

[revised manuscript text omitted]

---

## Author Comment (AC1)

**HESS Opinions: Chemical transport modeling in subsurface hydrological systems – Space, time, and the holy grail of "upscaling"**

**Brian Berkowitz**

**RC1**: Anonymous Referee #1

*Preliminary note:* I like this opinion paper very much and rate it "very instructive", delivering a good piece of "food for thought" for both early career and advanced scientist and also an eye-opener for a number of more applied researchers and practitioners! The paper is rated as an "opinion" paper. As such I hope that is will provoke scientific debate and discussions and trigger other submission that critically dispute the opinion provided by the author. For that cause, I limit my review to aspects I think need to be clarified.

Response: The Author appreciates the reviewer's positive appraisal of the manuscript. The constructive comments are also most appreciated; they are addressed in detail below, and in the manuscript to be revised, where appropriate.

**General comments**

The term "holy grail" is a very fundamental claim. Yet, the scientific perspective on flow and transport is rather specific addressing primarily the domain heterogeneity with respect to porosity, hydraulic conductivity and geochemical properties (limited to partitioning and sorption).

Response: The term "holy grail" can have different connotations to different readers. It is used here in the context of the flow/transport perspective of the manuscript, as noted by the Reviewer. Done: In light of the comment on this point by both Reviewers, the text in the revised manuscript will be modified by using quotation marks in the title and first use of the term in the text. Also, on first use of the term, a clarification of the intended meaning of the term in the context of the discussion will be provided.

There are important other aspects, still unresolved and most frequently ignored in literature, of fluid flow and chemical transport conceptualisation, theory, and modelling. Among others, these include the role of temporally and spatially changing fluid (water) properties like the viscosity and surface tension, or spatially and temporally changing properties of the immobile surface like mechanical or wetting properties, the role of mobile interfaces, or the transport of other materials than solutes for chemical transport. Importance arise from the fact that these factors have different spatial and temporal heterogeneity patterns that are incompatible with the ones addressed in this opinion paper. A fundamental and rigorous theoretical analysis is lacking. At time being, it is unclear whether an "explicit accounting of temporal information" will be enough to provide a way out.

Response: The Reviewer points out that other factors, such as viscosity, surface tension, wettability and transport of materials, can influence modeling of fluid and chemical transport. The

Author agrees, but notes that these factors are generally significant for multiphase flows, non-Newtonian fluid flow, and other scenarios; they are less relevant to the scenarios of flow and transport considered in the manuscript.

The Reviewer suggests that these other factors have spatial and temporal heterogeneity patterns that are incompatible with the ones addressed in this opinion paper. This point is certainly worth considering, but it is beyond the scope of the manuscript.

Given that the intent and scope of the manuscript appear clear, and in the interests of not expanding the text even further, the Author prefers not to include this discussion in the manuscript.

**Done:** In the revised manuscript, a brief statement will be included noting that these additional factors that are beyond the scope of the manuscript.

This holds also for the statement (L57 and on) on the relative effectiveness of upscaling approaches for "fluid flow". The "success" is limited to situations with fluids having the fluid properties of "pure water". I am not aware of any study that addresses temporal and spatial variation of the fluid viscosity. Once viscosity is considered no longer constant, even taking into account temporal effects, will not be sufficient to correctly represent "chemical transport".

**Response:** Agreed, although the ubiquity of non-constant viscosity scenarios in general groundwater systems might be considered limited.

**Done:** In the revised manuscript, the L57 statement will specify that constant fluid viscosity is assumed.

So "upscaling" with respect to heterogeneity in porosity, conductivity or geochemical properties fortified by an explicit accounting of temporal information is, admittedly, a grand challenge, but at least from my point of view, not the "holy grail".

Even when the attempt "to develop and apply chemical transport equations at large (length) scales, based on measurements and model parameter values obtained at significantly smaller length scales" would be successful, we must suppliantly recognize that we are still very far away from a "holistic" explanation (understanding) of transport in natural systems like soils and aquifers across scales. Thus, I recommend to "tone done" the paper by taking out the "holy grail".

**Response:** As noted in a Comment/Response above, the term "holy grail" is used in a specific context in the manuscript. The Reviewer suggests here that we are still very far from a full explanation of transport across scales. This comment would appear to agree with the Author's perspective, and specific use of the "holy grail" term.

**Done:** In the revised manuscript, as noted previously, the text will be modified by using quotation marks in the title and first use of the "holy grail" term in the text. Also, on first use of the term, a clarification of the intended meaning of the term in the context of the discussion is provided.

At the very beginning, the paper should point out that the presented examples on "chemical transport" are limited to situation of chemical transport of inert, i.e., non-reactive solutes. Although the author touches "reactive transport" in section 3.3.2, his paper does not elaborate this case.

**Response:** Agreed.
**Done:** In the revised manuscript, this point will be stated explicitly.

Even though the paper is an "opinion paper", I recommend eliminate the „Disclaimer". I strongly recommend to add the respective literature and references. It is good scientific conduct and will help the non-experts to navigate and reproduce the authors opinion by mirroring those with the existing "philosophies".

**Response:** The original manuscript included 42 references citing a range of author groups. Moreover, the manuscript is an "Opinion", rather than a "Review", the latter of which would indeed necessarily include at least 100-200 papers to fully survey and support consideration of all of the concepts and perspectives contained here.
**Done:** In the revised manuscript, and in light of the other reviewer's recommendation, ~20 additional citations will be included that expand on CTRW and other (non-time-centered) approaches ("philosophies"). The Author prefers to retain the "Disclaimer" in the Introduction, as the manuscript is not developed as a comprehensive review that surveys hundreds of papers.

Line 109 and section 1.3 Approach – Outline.: A graphical sketch of the hierarchy would help improve the perceivability.

**Response:** It is not clear to the Author how to present the hierarchy in graphical form, nor that doing so would add clarity. It appears that the text is sufficient to introduce the approach that is developed in the text that follows.

L116 and further: I recommend to detail what is meant "by measurements at similar scales of interest" and what type of "observational techniques" could/should be employed. It remains at this point quite unclear how this may be achieved in practice in the field. While structural features governing the permeability can eventually be measured at the same scale by, e.g., non-invasive geophysical methods, this is definitely not possible for the "geochemical heterogeneity" that controls retention and release. Most of the information on "reactive chemical transport" at the field scale regional scale has still to be delineated from - integrating - well measurements.

**Response:** Agreed.
**Done:** In the revised manuscript, these important clarifications will be introduced in a straightforward manner.

To a certain degree I feel that the examples used to illustrate the "opinion" are somewhat inconsistent. The paper motivates with natural porous media like "soil layers" and "subsurface geologic formations" and aquifers (see Abstract; L37-38 ). However, e.g. section 2.1 and Figure 1 are far from an even stochastic correct representation of natural subsurface geologic formations. Are the presented effects to expected as relevant or significant if more realistic "permeability" domains are considered?

**Response:** The Author understands the Reviewer's point. Figure 1 is an introductory, first example that focuses specifically on *pore-scale* flow behavior. There is no claim that this figure represents soil or aquifers, nor a stochastic representation of them.
**Done:** In the revised manuscript, text will be introduced to clarify the purpose of Figure 1 and the discussion associated with it, and to place it in perspective relative to the text that subsequently that focuses on soils and subsurface geological formations.

**Specific comments**

I recommend to reduce the relative clauses and commenting statements (e.g., the parenthesis) as these interrupt the "reading flow".

**Response:** Agreed.
**Done:** In the revised manuscript, the text will be modified throughout to improve the flow.

It is a single author paper. So, I recommend that the author refers to this by als writing "I" rather than "we".

**Response:** The use of "I" and "we" remains a subject of debate and writing style. Many statements are accepted by the community, so that "we" is in many contexts "more correct". Moreover, some statements are made based on results of multi-author studies, which again suggests that "we" is preferred, giving credit others who contributed to the cited paper(s). The Author therefore prefers to retain use of "we" throughout the manuscript.

Section 2 Fluid Flow: I am missing more recent approaches to resemble "connectivity" that are based on topology.

**Response:** OK.
**Done:** In the revised manuscript, brief mention of topology-based connectivity with citations will be included.

L496-497: Heterogeneities exist at all scales. Yet, AFM allows the resolution of small scale heterogeneities at resolutions; that prohibit to consider water still as a continuum fluid phase and the molecular properties of water govern dynamics and interactions with the porous medium. So, I suggest to eliminate the AFM example.

**Response:** OK. AFM was included only parenthetically.
**Done:** In the revised manuscript, this example will be deleted.

L645-647: Of course, at time being, a very detailed  - in the sense of resolution  - knowledge (measurement) on the spatial variability of the reactive surfaces is necessary to reconstruct "reactive" chemical transport.

**Response:** Agreed, but the text here is not referring at all to "reactive" chemical transport, which is mentioned later in the text.
**Done:** In the revised manuscript, in line with a Comment/Response above, clarifications will be introduced when referring to conservative and reactive chemical transport scenarios.

**Minor comments**

Chapter 3: Chemical transport. One may wonder in as far the observable effects (given in figures remain relevant or significant once the transport of an even only slightly "reactive" component is considered.

**Response:** The figures shown in Chapter 3 relate to conservative tracers. Chemical reactions introduce additional effects on transport patterns.
**Done:** In the revised manuscript, clear distinction between conservative and reactive transport situations will be made, with reference to the figures and discussion throughout the Chapter 3.

Figures 4 and 5: To what extent might the "non-Fickian" behaviour be due to the fact that the chemically inertness of the dye is not correct (due to sligth sorption of the dye to chemical impurities of the quartz sand).

**Response:** These figures are discussed in detail in the paper from which they were reproduced. Dye sorption was negligible. The text introducing these figures states explicitly that the red dye was inert.

**Technical corrections (non exhaustive)**

**Response:** The Author thanks the Reviewer for catching these typos, which the Author unfortunately missed.
**Done:** In the revised manuscript, the Author will make these corrections and others that were found.

L37 and on: replace „soil layers" by „soil"
L73: "case" instead of "Case"
L160: erase "is"
L174: replace "to not" by "do not"
L323: second "Path" change upper to lower case

L327: erase second "cause" in "does not cause necessarily cause a potential"
L371: erase first "only" in "knowledge only of only the flow"
L393: replace "show" by "shown"
L813; replace "dues" by "due"
L836: add a comma to "too".
L1060-1064: Revise. This sentence is confusing.
L1080: replace "many" by "may"

---

## Author Comment (AC2)

**HESS Opinions: Chemical transport modeling in subsurface hydrological systems – Space, time, and the holy grail of "upscaling"**

**Brian Berkowitz**

**RC2**: Referee #2 - Review by John Selker

The article provides an engaging and well-presented personal perspective on the science of transport of materials in natural porous media.

**Response:** The Author appreciates that the Reviewer finds the manuscript engaging and well-presented. As an Opinion paper, it is of course intended to provide a personal perspective.

The article is well titled in using the word "opinion," in that it reads as how the author thinks about these problems rather than seeking to provide a compelling case for his perspectives. I understand that the author and journal may see value in presenting opinions, which is their choice, but I must admit that I would have far preferred to spend my time reading a scientific article which provided compelling evidence and a well-rounded treatment of the diverse perspectives found in the literature. The lack of reference to the prominent and relevant work of Benson and Le Borgne, among many others, indicate to the reader that this is not a treatment of what has been shown in the literature, but rather what is believed by the author based on his own observations. I am not sure what I can do with such a presentation which straddles presentation of an opinion (which could have been well achieved in a very few paragraphs) and demonstration of principles, which would need to view the science as a community process rather than an individual sport.

**Response:** The Author suggests that, depending on the topic, it can be hard to provide the desired compelling case in just a few paragraphs, which cannot provide a fully justified and well-argued perspective. So a choice must be made. By surveying structure, fluid flow, and then chemical transport situations in the manuscript, and providing demonstrations of principles, the author believes that a compelling case is indeed developed for the need to incorporate an effective accounting of "time" in chemical transport modeling, and for aspects of "upscaling".
The Reviewer states that he would prefer to read "a scientific article which provided compelling evidence and a well-rounded treatment of the diverse perspectives found in the literature"…and then he points out work by two specific researchers as an "indication" that the manuscript does not reflect "what has been shown in the literature". Actually, in accord with the discussion in the manuscript, the two researchers mentioned by the Reviewer employ CTRW and time-fractional advection-dispersion formulations, the latter of which are known limit cases of CTRW. But "relevant and prominent work" is provided also by many others (in alphabetical order, Bijeljic, Blunt, Carrera, Dentz, Edery, Geiger, Gorelick, Guadagnini, Haggerty, Hansen, Juanes, and Metzler, to name just 12). Significantly, the perspectives given in all of these studies are not "different" – in the sense that they all include explicit treatment of time, and the specific mathematical relations are closely related, as noted in the manuscript. The "Disclaimer" supports the Author's decision to try to limit the choice of citations, and notes *explicitly* that "This approach is taken with a clear recognition and respect for the body of literature that has driven our field forward

over the last decades…". The Author therefore takes strong exception to the Reviewer's implication that the manuscript does not "view the science as a community process rather than an individual sport." This is indeed the Author's intention.

**Done:** The original manuscript already included 42 references citing a range of author groups. In the revised manuscript, and in light of the other reviewer's recommendation, ~20 additional citations will be included that expand on CTRW and other (non-time-centered) approaches. The Author prefers to retain the "Disclaimer" in the Introduction, as the manuscript is not developed as a comprehensive review that surveys hundreds of papers.

Overall, I find the article more emphatic than convincing – I did not count the exclamation points, but suppose there are on the order of 25. To this reader this elicited a sense that the author was too closely affiliated with his ideas to remain objective. Cooler arguments based on a broader reading of the literature would have been more convincing.

**Response:** There are precisely 13 exclamation marks, one of which appears in the Acknowledgements. Use of exclamation marks is stylistic, like many other aspects of writing. While scientific writing tends to eschew their use, this Author believes their appearance is justified when making a point that is surprising or unexpected. The Author accepts the Reviewer's personal sense that exclamation mark use indicates undesirable emotion. At the same time, the Author notes that use of exclamation points was not intended to intimate that "the author was too closely affiliated with his ideas to remain objective", nor that the author was offering less than "cool arguments"…… Indeed, of the 13 exclamation marks, only one is in the context of a citation to one of the author's papers.

**Done:** In the revised manuscript, to avoid any misunderstandings, all but one of the exclamation marks will be replaced by periods.

I have not studied the goals of HESS in presenting such opinions. From the article we get the sense that there are tight page limits, which is fine. I believe that the article would be far more effective and balanced if it were just one page long – just state that due to the fundamental role of time in spreading processes, combined with the multi-scale heterogeneity of geological media, that extrapolation in either time or space beyond a factor of two is an unreasonable expectation.

**Response:** See the response above: The manuscript is in accordance with current HESS Opinion paper criteria. The reviewer suggests that the same case could have been made in a one-page statement. One cannot provide the desired "compelling case", with a fully justified and well-argued perspective, in just a few paragraphs.

While I am well aware that the Holy Grail concept represents a reference to the unattainable to the author, in modern parlance this phrase is frequently employed to represent a remote, but potentially eventually attainable, objective. Such is language that there are multiple interpretations of a phrase. If the author wishes to be well understood, I would recommend including his intended meaning immediately following the first use in the text.

**Response:** The term "holy grail" can indeed have different connotations to different readers. **Done:** In the revised manuscript, in light of the comment on this point by both Reviewers, the text will be modified by using quotation marks in the title and first use of the term in the text. Also, on first use of the term, a clarification of the intended meaning of the term in the context of the discussion is provided.

I provide many additional observations on the PDF of the paper (attached) which present significant concerns I have, but do not rise to discussion in this over-arching consideration of the work.

**Response:** The Reviewer's attached pdf file contains a number of annotations, all of which have been considered carefully. Many annotations appear not in accordance with HESS style (e.g., annotated edits for format), and other marked changes to wording incorrectly change the intended meaning of a sentence. Other annotations suggest opportunities for further, straightforward clarification, which have been addressed. **Done:** In the revised manuscript, text will be modified in several locations to address those annotations that motivate helpful clarification.

---

## Author Response (AR1)

To the Editor, Thom Bogaard:

All point-by-point responses detailed in the previously-uploaded "Responses to Reviewers", as they appear online in the public discussion, have been addressed in the revised manuscript. I have made every effort to address the review comments as fully as possible.

In particular:
(1) Essentially all straightforward comments and requests for clarification were addressed, with appropriate modification/addition of text. In addition, writing style and wording was modified throughout.
(2) Use of the term "holy grail" was clarified.
(3) An additional 27 references are now included, 3 of which were necessarily with me as a co-author, but 24 from entirely independent research groups. These references provide additional support to my assertions and statements, as well as additional breadth and depth to related research developments.

I believe the manuscript to be clearer and stronger as a result of the revision.

Thank you for your consideration.

Best wishes,
Brian (Berkowitz)

---

## Author Response (AR2)

**RESPONSE TO THE EDITOR**

**Editor decision: Publish subject to technical corrections**
by Thom Bogaard

**Comments to the author**:
Dear Brian

thanks for the revised paper. I enjoyed reading it. It is long, but lively written and I can visualize you presenting it. I agree with the way you addressed the two reviews. I particularly think there is a nice balance now between references to earlier work and not too many references as it is an opinion paper with a build up of arguments for how chemical transport modelling (research) should be done (or what not should be done).
**Response:** I deeply appreciate the positive comments. We have made all noted technical corrections to the manuscript, as detailed below.

- A minor point is that I think the numerous use of "quotation marks" and italic font style and sometimes even combined, can be reduced. Often this in my opinion is adding a lot to the text or effective as writing style. So maybe review that.
**Response:** Done. The use of quotation marks and italics was reduced in several locations.

- Second, I think the figures would benefit from a visual scale in the corner of the figure. Even if it is approximated. (Fig 2 lower panel, fig 3 and 4)
**Response:** Done. Exact dimensions of domains in Figure 2, 3, and 4 are now provided.

- Minor observations:
Fig 3 caption: no doi reference needed here
**Response:** Done. Removed.

Fig 4 and 5 Caption: why (c) with permission if modified?
**Response:** Done. Corrected.

Section 5: Maybe one time remind the reader your work is on conservative chemical transport.
**Response:** Done. This is now noted in the first sentence of Section 5.

Lastly, thanks for this contribution to HESS, I am sure it will obtain quite some attention
**Response:** Many thanks!